# Trophic position of *Otodus megalodon* and great white sharks through time revealed by zinc isotopes

Jeremy McCormack[1,2✉], Michael L. Griffiths[3], Sora L. Kim[4], Kenshu Shimada[5,6], Molly Karnes[4], Harry Maisch[7], Sarah Pederzani[1], Nicolas Bourgon[1,8], Klervia Jaouen[1,9], Martin A. Becker[3], Niels Jöns[10], Guy Sisma-Ventura[11], Nicolas Straube[12], Jürgen Pollerspöck[13], Jean-Jacques Hublin[1,14], Robert A. Eagle[15] & Thomas Tütken[8]

Diet is a crucial trait of an animal's lifestyle and ecology. The trophic level of an organism indicates its functional position within an ecosystem and holds significance for its ecology and evolution. Here, we demonstrate the use of zinc isotopes ($\delta^{66}$Zn) to geochemically assess the trophic level in diverse extant and extinct sharks, including the Neogene megatooth shark (*Otodus megalodon*) and the great white shark (*Carcharodon carcharias*). We reveal that dietary $\delta^{66}$Zn signatures are preserved in fossil shark tooth enameloid over deep geologic time and are robust recorders of each species' trophic level. We observe significant $\delta^{66}$Zn differences among the *Otodus* and *Carcharodon* populations implying dietary shifts throughout the Neogene in both genera. Notably, Early Pliocene sympatric *C. carcharias* and *O. megalodon* appear to have occupied a similar mean trophic level, a finding that may hold clues to the extinction of the gigantic Neogene megatooth shark.

[1] Department of Human Evolution, Max Planck Institute for Evolutionary Anthropology, Leipzig, Germany. [2] Institute of Geosciences, Goethe-University Frankfurt, Frankfurt am Main, Germany. [3] Department of Environmental Science, William Paterson University, Wayne, NJ, USA. [4] Department of Life and Environmental Sciences, University of California Merced, Merced, CA, USA. [5] Department of Environmental Science and Studies and Department of Biological Sciences, DePaul University, Chicago, IL, USA. [6] Sternberg Museum of Natural History, Fort Hays State University, Hays, KS, USA. [7] Department of Marine and Earth Sciences, Florida Gulf Coast University, Fort Myers, FL, USA. [8] Institute for Geosciences, Johannes Gutenberg-University, Mainz, Germany. [9] Géosciences Environnement Toulouse, CNRS, Observatoire Midi Pyrénées, Toulouse, France. [10] Department of Geology, Mineralogy and Geophysics, Ruhr University Bochum, Bochum, Germany. [11] Israel Oceanographic & Limnological Research Institute, Haifa, Israel. [12] Department of Natural History, University Museum of Bergen, Bergen, Norway. [13] Bavarian State Collection of Zoology, Munich, Germany. [14] Chaire de Paléoanthropologie, Collège de France, Paris, France. [15] Institute of the Environment and Sustainability, Department of Atmospheric and Oceanic Sciences, University of California, Los Angeles, CA, USA. ✉email: mccormack@em.uni-frankfurt.de

Among the greatest challenges in palaeobiology is reconstructing ecological niches and the trophic hierarchy of extinct species, including apex and mesopredators like sharks. Diet plays an essential role in each species' evolution and extinction, where trophic interactions can stimulate competition while the loss of prey species can have detrimental effects on a species' survival. Nevertheless, decisively identifying the overall diet of any long-extinct animal is complex and often relies on anatomy-based inferences or taphonomic evidence, such as fossilised stomach content, bite marks, or identifiable remains in fossilised faeces[1–5]. However, these occurrences are rare in the fossil record, and while they provide a snapshot of predator-prey interactions, they do not necessarily reflect a species' diet over an extended period of time nor its trophic level and thereby functional position in an ecosystem.

Zinc is essential for living organisms and plays a crucial role in various biological processes, including bioapatite mineralisation[6]. Zinc in vertebrate tissues, including in the mineral phase of skeletal tissues (bioapatite), mainly comes from the diet[7,8] and its isotope ratios ($^{66}$Zn/$^{64}$Zn expressed as $\delta^{66}$Zn) are a useful trophic level proxy in mammals[9–13]. Noteworthy, studies on marine mammal $\delta^{66}$Zn values are very limited and were so far only performed on archaeological material[10,13]. Nevertheless, all studies to date indicate that $\delta^{66}$Zn values successively decrease with increasing trophic level[9–13] (Supplementary Note 1). This trophic, dietary Zn isotope signal is incorporated into bioapatite forming enamel or enameloid of teeth, a hard tissue, which is highly resistant to alteration and likely provides better long-term preservation than organic collagen-bound nitrogen[11], the most commonly used isotopic trophic tracer to date. However, there are no published reports on $\delta^{66}$Zn values of non-mammalian vertebrate bioapatite, except for enamel and bone of three African crocodilians[9], nor of any fossil teeth older than the Late Pleistocene[11,12].

Here, we first examine the zinc isotope systematics of modern shark and teleost fish bioapatite from various locations. Specifically, we explore bioapatite $\delta^{66}$Zn values of different tissues (enameloid, dentine, gill raker, and bone) with traditional diet-based bulk collagen carbon and nitrogen isotope ($\delta^{13}C_{coll}$, $\delta^{15}N_{coll}$) values in 20 extant species, including control fed aquarium and pisciculture fish. We consider variation due to key factors, such as diet, habitat use, and baseline variability. We then investigate the deep-time preservation potential of this proxy—i.e., evaluating potential diagenetic versus biological influences on fossil enameloid $\delta^{66}$Zn signals—by sampling different fossil tissues (enameloid versus dentine) and performing electron microprobe Zn mapping. We further compare tooth phosphate oxygen isotope ($\delta^{18}O_P$) values in a subset of fossil samples to infer species-specific relative palaeowater temperature preferences. Finally, comparing absolute and relative differences among enameloid $\delta^{66}$Zn values of extant and extinct species allows us to examine the foraging ecology of various extinct species including *Otodus megalodon*.

*Otodus megalodon* is the last chronospecies of the Paleocene–Pliocene megatooth shark (*Otodus*) lineage[14,15] and one of the largest carnivores to have ever lived[16]. Diet and resource competition have been discussed as possible drivers of its evolution and extinction[4,15], but its abrupt disappearance in the fossil record remains an enigma. In this study, we use zinc isotopes to compare the dietary ecology of two successive Miocene–Pliocene chronospecies in the megatooth sharks' lineage, *O. chubutensis* and *O. megalodon*, to that of the Pliocene–extant great white shark (*Carcharodon carcharias*). If these apex predators fed at similarly high trophic levels, there is the possibility of resource competition. Further, the extent of trophic variation within and between localities could yield important clues to trophic dynamics, ecological plasticity, and evolutionary "success."

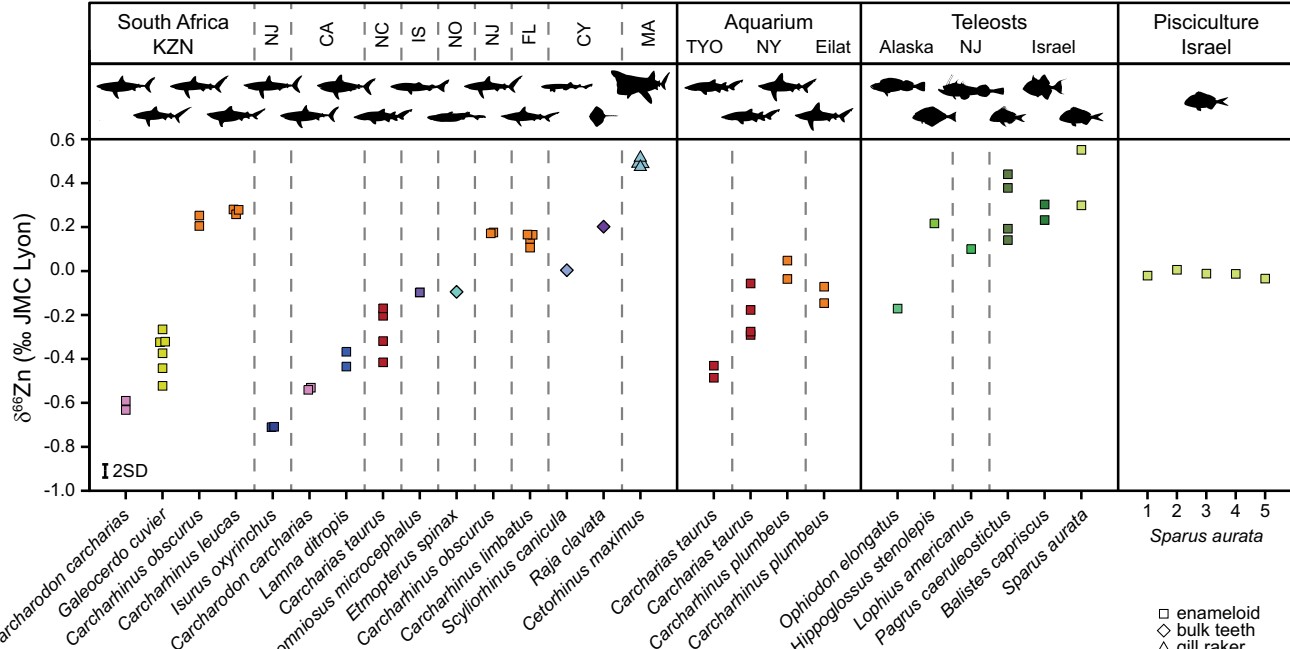

**Fig. 1 Zinc isotope ($\delta^{66}$Zn) composition of extant elasmobranch and teleost fish teeth and gill raker.** Specimens come from off the coast of KwaZulu-Natal (KZN) South Africa, New Jersey (NJ), California (CA), North Carolina (NC), Iceland (IS), Norway (NO), Florida (FL), Cyprus (CY), Massachusetts (MA), Alaska and Israel. Aquarium sharks are from the New York (NY) and Tokyo (TYO) Aquariums and the Eilat (Israel) Underwater Observatory Park. Pisciculture *S. aurata* individuals are numbered and plotted individually to visualise the homogeneity among control-fed individuals compared to wild elasmobranchs and teleosts. Silhouettes are not to scale. Measurement uncertainty is indicated at the 2 SD level. Samples are colour-coded following their genus, regardless of locality. Source data are provided as a Source Data file.

## Results and Discussion

Our study reveals zinc isotopes to be a promising trophic indicator in sharks and other fishes in general, similar to previous studies featuring both terrestrial and marine mammals[9–13]. We analysed the Zn isotope values for extant sharks spanning captive/aquarium and wild individuals from various localities and found close correspondence with their respective trophic level. Further, Zn concentration and isotope composition suggest preservation of this biological signal in fossil specimens with little diagenetic alteration. A survey of fossil shark teeth spanning the Miocene–Pliocene reveal similar $\delta^{66}$Zn values and variation as found in related (e.g., congeneric) extant sharks with similar dentition and ecology. Our $\delta^{66}$Zn results indicate high trophic levels for *Otodus* and perhaps a trophic change in *C. carcharias*, the great white shark.

**Zinc isotopes in extant sharks and teleosts**. As with mammals[9–13], bioapatite $\delta^{66}$Zn values in wild extant elasmobranchs and teleosts show overall lower values with increasing trophic level (Fig. 1; Supplementary Fig. 1). Both $\delta^{66}$Zn and $\delta^{15}$N$_{coll}$ correlate with FishBase[17] trophic levels, despite differences in species' geographic origin and tissue types sampled (Spearman's correlation $r = -0.87$, $p = 5.65$E$-16$, $n = 48$ and $r = +0.42$, $p = 1.47$E$-5$, $n = 40$, respectively). There is no statistically significant relationship between bioapatite $\delta^{66}$Zn values and $\delta^{13}$C$_{coll}$, but there is one between wild fish $\delta^{66}$Zn and $\delta^{15}$N$_{coll}$ values from the same tooth or individual ($R^2 = 0.28$, $p = 6.89$E$-4$, $n = 38$; Supplementary Fig. 2). Both proxies thus generally reflect trophic levels. Large apex predatory sharks (e.g., *Carcharodon carcharias*, *Isurus oxyrinchus*, and *Lamna ditropis*) have significantly more negative $\delta^{66}$Zn values than lower trophic level teleosts and the plankton-feeding basking shark (*Cetorhinus maximus*). In particular, the shortfin mako shark (*I. oxyrinchus*) and great white shark (*Carcharodon carcharias*), both apex predators[18,19], have much lower $\delta^{66}$Zn values than in any previously recorded extant vertebrate species (enameloid up to –0.71 and –0.63‰, respectively). These low $\delta^{66}$Zn values are likely due to the larger number of trophic levels in the marine ecosystem than in terrestrial food webs (terrestrial mammal enamel lies typically between 0 and +1.6‰[9,11]) and perhaps differences between marine and terrestrial Zn isotope baselines.

Absolute enameloid $\delta^{66}$Zn values vary by up to 1.26‰ among the extant species analysed from various oceanographic areas (Fig. 1). Our results also demonstrate large variability in enameloid $\delta^{66}$Zn values among extant sharks within the same region; for example, there is a 0.88‰ difference between mean values of *Carcharodon carcharias* and the bull shark (*Carcharhinus leucas*; Fig. 1) from KwaZulu-Natal (South Africa, KZN). In contrast, enameloid $\delta^{66}$Zn from the same species (e.g., *Carcharodon carcharias*, *Carcharhinus obscurus*) demonstrate a low isotopic variability, independent of geographic location (Fig. 1).

We observe uniformity in the enameloid $\delta^{66}$Zn values of five gilt-head bream (*Sparus aurata*) individuals fed on a controlled fish pellet diet in pisciculture cages located offshore of Central Israel, with values within the measurement uncertainty of each other (–0.01 ± 0.01‰). As with $\delta^{66}$Zn, the $\delta^{15}$N$_{coll}$ and $\delta^{13}$C$_{coll}$ values are distinct from those of wild teleost individuals caught nearby in Haifa Bay, reflecting the artificial pelleted diet of the pisciculture individuals (Fig. 1, Supplementary Figs. 3, 4). Strongly contrasting the homogenous control fed *S. aurata* $\delta^{66}$Zn values, we observe a higher $\delta^{66}$Zn variability among (and even within) wild and aquaria elasmobranch individuals (fed with wild-caught fish and cephalopods). For instance, two teeth of a single tiger shark (*Galeocerdo cuvier*) individual (–0.52 and –0.27‰, Fig. 1) have a variability higher than the total variability

among the three KZN *G. cuvier* individuals. *Galeocerdo cuvier* is well known for its highly opportunistic prey selection[20]. Therefore, the $\delta^{66}$Zn value of bioapatite is likely highly responsive to an individual's diet at the time of tissue formation, and as shark teeth form and replace continuously, enameloid $\delta^{66}$Zn values can vary among teeth of a single individual. Thus, although fish can absorb Zn via their gills, waterborne Zn absorption appears to have a negligible effect on elasmobranch tooth $\delta^{66}$Zn values, in line with Zn incorporated into soft and skeletal tissues in natural environments being predominantly derived from dietary gastrointestinal uptake[7,8].

*Carcharhinus* enameloid $\delta^{66}$Zn values are high relative to sharks with similar bulk $\delta^{15}$N$_{coll}$ values, which contrary to the here analysed *Carcharhinus* species more regularly consume pelagic prey offshore, oceanic and on the continental shelf (e.g., *Galeocerdo cuvier*)[21]. This discrepancy may relate to *Carcharhinus* species inhabiting neritic waters where they feed primarily on demersal/benthic, freshwater-brackish-coastal prey[22–25]. While the diet of KZN *G. cuvier* and *Carcharodon carcharias* can also include reef-associated or demersal prey, pelagic organisms are typically more important by mass, especially in adult individuals[20,26]. Zinc isotope variability among marine organisms and their tissues is largely unknown, currently limiting our ability to identify specific food items based on shark enameloid $\delta^{66}$Zn values beyond generally observed trophic level effects. Whether higher *Carcharhinus* enameloid $\delta^{66}$Zn values relate to specific prey species (and trophic level) or general differences in basal organic matter source between a primarily neritic food web compared to a more open marine pelagic food web remains unclear (Supplementary Discussion 1). However, we observe no difference in $\delta^{13}$C$_{coll}$ values that imply a more terrestrial carbon signal in the KZN *Carcharhinus* species relative to sympatric species, arguing against differences in the basal organic matter source amongst the KZN shark species (Supplementary Fig. 3).

A previous study on Arctic marine mammal bones suggested a higher geographic independence of $\delta^{66}$Zn values from baseline variability compared to $\delta^{13}$C$_{coll}$ and $\delta^{15}$N$_{coll}$ values[13]. Likewise, fish taxa with similar diet composition, habitat use and/or trophic level, have a similar range of bioapatite $\delta^{66}$Zn values regardless of their geographic locality (Fig. 1), indicating that $\delta^{66}$Zn may allow worldwide dietary and trophic level comparability with limited marine baseline variation. Further studies will need to expand our knowledge on $\delta^{66}$Zn variability in extant marine vertebrates as well as the effects of baseline on marine vertebrate enameloid $\delta^{66}$Zn values, especially compared to dentine $\delta^{13}$C$_{coll}$ and $\delta^{15}$N$_{coll}$ values. Nevertheless, the high taxa-specific and perhaps baseline-independent $\delta^{66}$Zn values suggest $\delta^{66}$Zn is an independent indicator of trophic level and an asset for present and past food web reconstructions in the marine realm.

**Deep-time zinc isotope preservation in fossil enameloid**. Fossil enameloid has $\delta^{66}$Zn values and Zn concentrations ([Zn]) in the range of extant elasmobranch species, arguing against significant diagenetic modification (Figs. 2, 3, Supplementary Fig. 5). Fossil shark teeth examined herein are from Germany, Malta, Japan, North Carolina (USA) and Florida (USA) covering the Early Miocene (Burdigalian, 20.4–16.0 Ma), Miocene-Pliocene transition (Messinian-Zanclean boundary, ca. 5.3 Ma), and the Early Pliocene (Zanclean, 5.3–3.6 Ma; Fig. 2; Supplementary Note 2). Importantly, extant and fossil elasmobranch enameloid $\delta^{66}$Zn values (–0.71 to +0.28‰ and –0.83 to +0.27‰, respectively) differ from: (1) previously reported values of terrestrial mammal enamel (0 to +1.6‰);[9,11] and (2) sedimentary carbonate $\delta^{66}$Zn values of the fossil sites (+0.34 to +0.49‰, Supplementary Fig. 6,

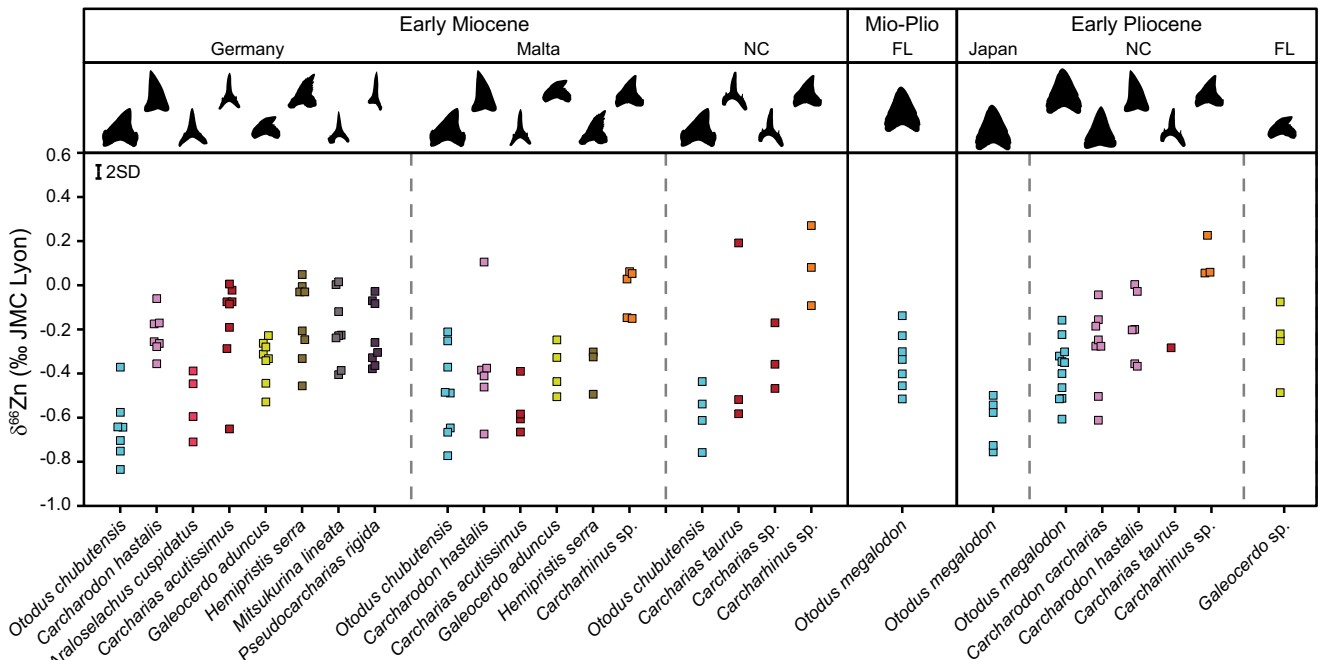

**Fig. 2 Zinc isotope (δ⁶⁶Zn) composition of fossil elasmobranch enameloid.** Teeth are from the Early Pliocene of Japan, North Carolina (NC) and Florida (FL), Pliocene to Miocene transition of Florida, the Early Miocene of North Carolina, Germany and Malta. For more details on the sample background, see Supplementary Data 1, Supplementary Note 2. Silhouettes are not to scale. Measurement uncertainty is indicated at the 2 SD level. Samples are colour-coded following their genus, regardless of locality. Source data are provided as a Source Data file.

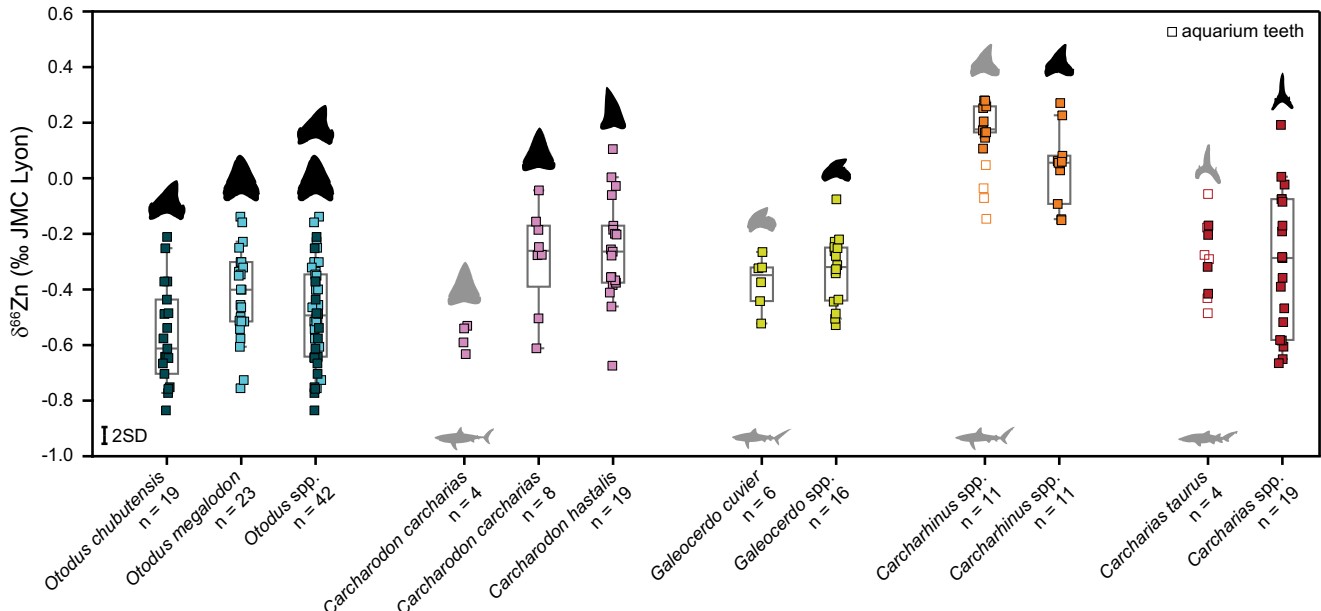

**Fig. 3 Zinc isotope (δ⁶⁶Zn) composition of fossil and extant elasmobranch enameloid of selected taxa combined from Figs. 1 and 2.** Fossil teeth are from multiple locations and ages. Grey silhouettes indicate extant teeth. The boxes for *n* > 5 represent the 25th–75th percentiles (with the median as a horizontal line) and the whiskers show the 10th–90th percentiles. Box plots (and *n*) do not include aquarium teeth (open squares). *Otodus* spp. includes all *O. chubutensis* (dark blue) and *O. megalodon* teeth (light blue) analysed and samples are otherwise colour-coded following their genus. Silhouettes are not to scale. For more details on the samples, see Figs. 1 and 2 and Supplementary Data 1. Source data are provided as a Source Data file. Measurement uncertainty is indicated at the 2 SD level.

Supplementary Table 1). These differences support a preserved biological signal in fossil enameloid.

We observe the same within tooth Zn spatial concentration pattern in extant and Miocene tiger shark (*Galeocerdo* spp.) teeth, with Zn being more enriched in the outer enameloid than close to the enameloid-dentine junction. If significant diagenetic Zn

exchange had occurred throughout the enameloid, this original Zn concentration pattern would not be preserved in the fossil tooth (Supplementary Fig. 7). Additionally, both extant and fossil shark enameloid show the same variation in [Zn] according to their taxonomy, with carcharhiniforms generally having higher [Zn] than lamniforms (Supplementary Fig. 5), again arguing

against significant diagenetic enameloid Zn exchange. For the European Miocene sites, $\delta^{18}O_P$ analyses were also conducted on a subset of teeth, where their enameloid appears to be generally well-preserved as suggested by $\delta^{18}O_P$ values demonstrating species-specific relative in-vivo temperature ranges as expected compared to the habitat use of equivalent modern species[27] (Supplementary Fig. 8).

To discern the effects of diagenetic Zn alteration, we compare visually pristine appearing enameloid with areas sampled along fractures and dentine of the same tooth (Supplementary Figs. 6 and 9, Supplementary Table 2). Our results demonstrate that the diagenetic Zn exchange in fractured enameloid leads to higher $\delta^{66}Zn$ values than in the pristine enameloid of the same tooth, whereas we observe no differences in enameloid $\delta^{66}Zn$ profiles of modern teeth (Supplementary Figs. 9 and 10, Supplementary Table 3). Likewise, the diagenetically more susceptible fossil dentine shows higher $\delta^{66}Zn$ values as reflected by a significantly higher and more variable dentine-enamel $\delta^{66}Zn$ offset (+0.78 ± 0.33‰, $n = 13$) than observed for extant teeth (+0.22 ± 0.1‰, $n = 23$; Supplementary Discussion 2, Supplementary Figs. 6 and 11). For the fossil enameloid shown here, in-vivo $\delta^{66}Zn$ values must be at least as low as their current values, indicating limited to no alteration. Consequently, $\delta^{66}Zn$ analysis of fossil enameloid can enable deep-time dietary reconstructions.

The homogeneity in $\delta^{66}Zn$ values for the same species or genera independent of locality and geological age is a remarkable observation (Figs. 2, 3), not only limiting the likelihood of Zn diagenetic alteration but also arguing for minimal variability in habitat-specific food web baselines or a strong homogenisation of $\delta^{66}Zn$ values at low trophic levels. There are still some limitations, such as the absence of reported $\delta^{66}Zn$ values of marine non-mammalian vertebrates for comparison outside this study, the limited sample size for some species, and uncertainties regarding Zn isotope baseline variability. However, our extensive $\delta^{66}Zn$ dataset includes not only multiple species from different localities and periods with distinct differences in dietary Zn uptake among extinct elasmobranch species, but also direct overlap in extant and fossil $\delta^{66}Zn$ values of the same genus and/or lifestyle. This spatial and temporal coherence suggests that it may be possible to use the same interpretative framework on extant and fossil elasmobranch assemblages globally (Figs. 1, 2, 3), and our remaining discussion is based on this assumption.

**Zinc isotopes and ecology of Miocene-Pliocene sharks.** Absolute and relative $\delta^{66}Zn$ values among some taxonomic groups show no statistical variation with geologic age and locality (e.g., *Carcharias* spp., *Galeocerdo* spp.), indicating relatively stable trophic levels and ecological niches throughout time and space. For example, most extinct elasmobranchs with a slender tearing, grasping tooth morphology (e.g., *Carcharias*) have $\delta^{66}Zn$ values that can be directly compared to modern equivalents (e.g., *Carcharias taurus*, *Isurus oxyrinchus*, *Lamna ditropis*). This type of dentition and corresponding tooth morphology are adapted to restrain small, active prey—like fish and cephalopods[28–30]. However, there are differences among the $\delta^{66}Zn$ values for these types of elasmobranchs within the Early Miocene of Germany, with *Mitsukurina lineata* and *Pseudocarcharias rigida* having higher mean $\delta^{66}Zn$ values compared to *Araloselachus cuspidatus* (Fig. 2). Indeed, post hoc Tukey pairwise comparisons draw out *A. cuspidatus* as distinct from most species for the Germany (Early Miocene) assemblage, including those with a similar grasping tooth morphology (Supplementary Table 4). Our $\delta^{66}Zn$ values indicate that *A. cuspidatus* was likely a higher trophic level piscivore than *M. lineata* and *P. rigida*, supported by the larger tooth size of *A. cuspidatus*.

Zinc isotope values within the *Galeocerdo* lineage show no statistical variability with age nor locality, suggesting tiger sharks occupied a similar trophic level and ecological role in the marine ecosystem since at least the Early Miocene (Fig. 3). Notably, our results imply that the increase in body size from *G. aduncus* to the modern *G. cuvier* did not change its overall trophic level, which is in line with the highly similar tooth morphology between the two species[31].

For *Carcharhinus*, the Early Miocene Malta assemblage is drawn out as statistically different from extant wild *Carcharhinus* spp. (Supplementary Table 5). Still, *Carcharhinus* spp. in both extant and fossil assemblages always have higher mean $\delta^{66}Zn$ values than other sympatric predatory sharks and are drawn out as statistically different from sympatric shark species in each fossil assemblage (Figs. 2, 3, Supplementary Tables 6–8). Based on similarities in tooth morphology and $\delta^{66}Zn$ values among extant and extinct *Carcharhinus* spp., we suggest that extinct taxa also primarily occupied a neritic-coastal habitat feeding upon demersal-benthic prey[22–25]. For the *Carcharhinus* teeth from Malta, this interpretation is supported by lower $\delta^{18}O_P$ values than sympatric species, indicating a higher water temperature or lower salinity: i.e., a shallow and/or brackish water habitat (Supplementary Fig. 8). Consequently, the uniformly higher $\delta^{66}Zn$ values of extant and fossil *Carcharhinus* spp. indicate the consumption of food items distinct from other measured sympatric species already during the Early Miocene and Early Pliocene.

Absolute $\delta^{66}Zn$ values for *Otodus* spp., along with values relative to sympatric species, indicate megatooth sharks were apex predators feeding at a very high trophic level (Figs. 2, 3). In all Early Miocene assemblages, mean *O. chubutensis* $\delta^{66}Zn$ values are among the lowest compared to sympatric species, including the lowest bioapatite $\delta^{66}Zn$ value measured to date (–0.83‰). Mean *O. chubutensis* $\delta^{66}Zn$ values are as low as extant *Carcharodon carcharias* (respectively, –0.57 ± 0.18‰, $n = 19$ and –0.57 ± 0.05‰, $n = 4$). Noteworthy, Games-Howell pairwise comparisons indicate the lower extant *C. carcharias* $\delta^{66}Zn$ values as distinct from most fossil *Carcharodon* populations, possibly indicating a dietary shift in the *Carcharodon* lineage (Supplementary Table 9). Early Pliocene values from *O. megalodon* from Japan also demonstrate very low mean $\delta^{66}Zn$ values (–0.62 ± 0.11‰, $n = 5$) that are statistically different from the Atlantic *O. megalodon* populations sampled from Florida and North Carolina, which have higher mean $\delta^{66}Zn$ values (respectively, –0.34 ± 0.11‰, $n = 11$; –0.38 ± 0.11‰, $n = 7$; Fig. 2, Supplementary Table 10).

Possible explanations for the observed spatial and temporal variability in *Otodus* and *Carcharodon* $\delta^{66}Zn$ values in our study are differences in prey consumption (and trophic level) or baseline variation. Additionally, we cannot rule out other factors such as interpretive limitations due to sample sizes. For example, extant *C. carcharias* can exhibit some degree of dietary individuality[32], yet we only have $\delta^{66}Zn$ data from two individuals (4 teeth) from two localities. Still, the low $\delta^{66}Zn$ values in both extant *C. carcharias* compared to other extant sharks is in line with the generally high trophic level estimates of this species[18]. Particularly for *O. megalodon* from Japan where we have only one species analysed, we cannot exclude the possibility of either differences in $\delta^{66}Zn$ baseline or regionally different prey species. However, the absence of significant $\delta^{66}Zn$ differences within many taxa amongst locations and geological ages implies negligible differences in $\delta^{66}Zn$ food web baselines (Figs. 2, 3). Therefore, the observed spatial and temporal variability in $\delta^{66}Zn$ values likely demonstrates true dietary differences amongst *Otodus* and *Carcharodon* populations both geographically and temporally, with important implications for each species' feeding ecology and evolution both on a local and global scale.

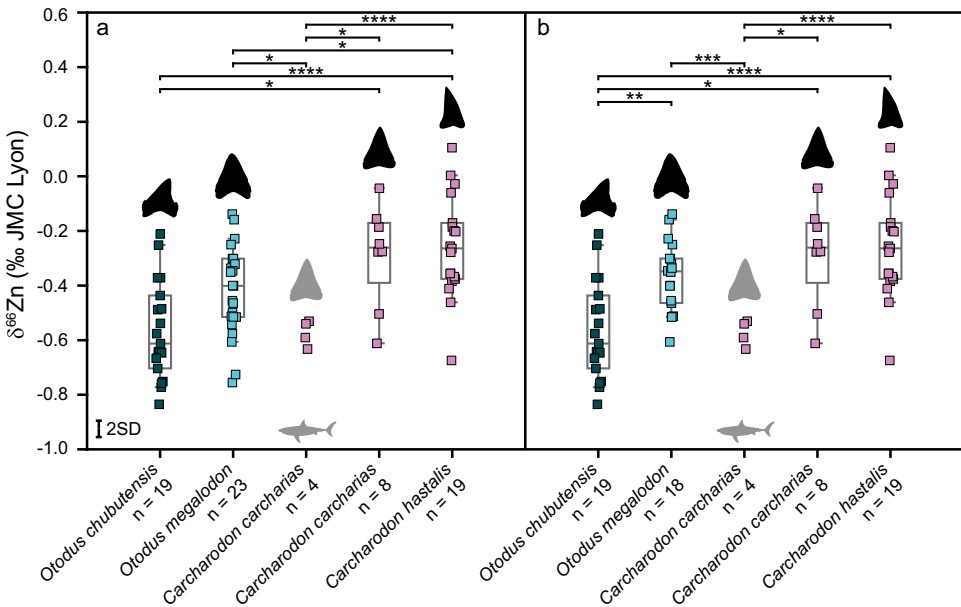

**Fig. 4 Results from post-hoc Games-Howell pairwise comparisons of δ⁶⁶Zn values of enameloid from fossil and extant *Otodus* spp. and *Carcharodon* spp.** All assemblages and ages are combined for a given species, except for extant *C. carcharias*. Extant *C. carcharias* teeth are indicated with grey silhouettes. **a** Includes all *O. megalodon* populations, whereas (**b**) excludes the Japanese (Pacific) population, focusing on Atlantic and Tethys/Paratethys populations only. The boxes for $n > 5$ represent the 25th–75th percentiles (with the median as a horizontal line), and the whiskers show the 10th–90th percentiles. Significance level is indicated by "*" ($p$ value < 0.05), "**" ($p$ value < 0.005), "***" ($p$ value < 0.0005) and "****" ($p$ value < 0.00005). Measurement uncertainty is indicated at the 2 SD level. See also Supplementary Tables 11, 12. *Otodus chubutensis* (dark blue) and *O. megalodon* teeth (light blue) are coloured separately. All other samples are colour-coded following their genus. Source data are provided as a Source Data file. Silhouettes are not to scale.

*Otodus* and *Carcharodon* in the Early Miocene are represented by *O. chubutensis* and *C. hastalis*, respectively, with statistically significant higher mean δ⁶⁶Zn values from the latter (Figs. 3, 4, Supplementary Tables 11, 12). The mean δ⁶⁶Zn value for all *O. chubutensis* is the lowest of all mean values recorded in our fossil shark dataset (Fig. 3), suggesting that *O. chubutensis* could occupy a higher trophic position than *C. hastalis*. Importantly, differences between δ⁶⁶Zn values of *O. chubutensis* and *C. hastalis* do not appear related to a different ratio of juveniles to adults in either species, as our results do not record an ontogenetic diet shift (Supplementary Fig. 12). We observe no correlation between the total body length of *Otodus* spp., *Carcharodon* spp. and their respective δ⁶⁶Zn values (Supplementary Fig. 12), likely, because each examined specimen had already surpassed the body size for which ontogenetic dietary shifts, if any, occur.

When including only *Otodus* spp. from the Atlantic and Paratethys/Tethys regions, we observe a statistically significant difference between *O. chubutensis* and *O. megalodon* (Supplementary Table 12, Fig. 4b). During the Early Pliocene, the *Otodus* lineage represented by *O. megalodon* shows a considerable increase in the mean δ⁶⁶Zn value for the Atlantic populations, hinting at a reduced trophic position for the megatooth shark lineage in the Atlantic. At the same time, the Early Pliocene *C. carcharias* remains at the same trophic level as *C. hastalis* (Figs. 2–4, Supplementary Tables 11, 12). Although the extant sample size is limited, our results are intriguing because the mean δ⁶⁶Zn value for extant *C. carcharias* places it at a trophic level that would be higher than the Atlantic Early Pliocene *O. megalodon* (Figs. 3, 4, Supplementary Table 11, 12).

Extant *Carcharodon carcharias* is a predatory shark whereby larger individuals regularly feed on high trophic level marine mammals[33]. Although Neogene *Carcharodon* and *Otodus* were likely opportunistic in their prey selection similar to many extant apex predatory sharks[33], fossil evidence of bite marks suggests

that both taxa fed largely on marine mammals such as cetaceans (mysticetes and odontocetes) and pinnipeds[1,2,4,34–38]. However, in the majority of cases, it remains unclear if these feeding events on mammals document active hunting or scavenging and how important each prey taxa were to their overall diet. Early Pliocene *C. carcharias* and *O. megalodon* δ⁶⁶Zn data suggest that lower trophic level mammal prey such as mysticetes (and perhaps herbivorous sirenians) may have been an important food item for both species. Mysticetes are filter-feeders and likely to have higher tissue δ⁶⁶Zn values than piscivorous odontocetes or pinnipeds, similar to the higher δ⁶⁶Zn values in the plankton-feeding extant *Cetorhinus maximus* compared to piscivorous sharks (Fig. 1). Bite marks on Late Miocene–Early Pliocene mysticetes bones from both *Carcharodon carcharias* and *O. megalodon*[1,4,34,38] corroborate at least occasional feeding events.

Now extinct small- and medium-sized mysticetes (e.g., Cetotheriidae and various small-sized Balaenidae and Balaenopteridae) were abundant during the Early Pliocene[39,40] and were thus available as prey for large sharks, i.e., *Otodus megalodon*[4] and *Carcharodon carcharias*[1]. In contrast, Early Miocene cetacean fossils are dominated by toothed cetaceans, where the Early Miocene European and North American sites sampled in this study lack any mysticete remains[41–44]. The Early Miocene *Otodus* (and modern *C. carcharias*) lower δ⁶⁶Zn values (higher trophic level) may partly be related to the lack of lower trophic level mammals (e.g., mysticetes) available as prey. Mysticetes became more abundant following a diversity plateau during the mid-Miocene[39,45]. Subsequently mysticetes remains become more prominent in the Late Miocene to Early Pliocene fossil assemblages from North Carolina and Florida studied herein[43,44], where mysticetes were, together with other mammals (e.g., odontocetes), possibly preyed upon by *O. megalodon* and *C. carcharias*.

For the Early Pliocene of North Carolina, where we have δ⁶⁶Zn values for both *Otodus megalodon* and *Carcharodon carcharias*,

our results suggest largely overlapping trophic levels for both species. Feeding at the same trophic level does not necessarily imply direct dietary competition, as both species could have specialised on different prey with similar trophic levels. However, at least some overlap in food items between both species is likely, as also indicated by fossil bite marks[1,4,34,38]. Extant predatory sharks typically feed on a wide range of food items[33], and there is evidence for generalist feeding, as well as, in some cases, specialisation at lower trophic levels for extant *C. carcharias*[32]. Higher dietary individuality and the opportunistic nature of apex predators are possible explanations for the range of $\delta^{66}Zn$ values observed in both species (–0.61 to –0.04‰ in Pliocene North Carolina).

The extinction of *Otodus megalodon* could have been caused by multiple, compounding environmental and ecological factors[46,47], including climate change and thermal limitations[48], the collapse of prey populations[4] and resource competition with *Carcharodon carcharias*[15] and possibly other taxa not examined here (e.g., carnivorous odontocetes). The $\delta^{66}Zn$ results presented here indicate the potential of trophic change, where we find evidence for a decrease in the mean trophic position from *O. chubutensis* to *O. megalodon* in the Atlantic and an increase in trophic position for *C. carcharias* from the Early Pliocene to its extant form. If these trophic dynamics are accurate, then there is a possibility for the competition of dietary resources between these two shark lineages[15]. Our results also support the hypothesis of *Otodus* size-driven co-evolution and co-extinction with mysticetes[4], indicated, at least for the Atlantic assemblage, by a shift towards lower trophic level prey from the Early Miocene to the Early Pliocene within the *Otodus* lineage. In general, our study demonstrates $\delta^{66}Zn$ to be a powerful, promising tool to investigate the trophic ecology, diet, evolution, and extinction of fossil marine vertebrates.

## Methods

**Material**. All specimens used in this study were collected legally and ethically, and are housed in the following institutions that also permitted us to conduct destructive sampling for the purpose of this study through loan agreements: Calvert Marine Museum, Solomons, Maryland, USA; Osteological Collection, Institute of Geosciences, Johannes Gutenberg-University, Mainz, Germany; Natural History Museum of Los Angeles County, California, USA; Massachusetts Natural History Collections, University of Massachusetts, Amherst, Massachusetts, USA; University Museum of Bergen, Bergen, Norway; and the Field Museum of Natural History, Chicago, Illinois, USA (Supplementary Data 1). We analysed enameloid, dentine and bone $\delta^{66}Zn$ values of extant elasmobranch and teleost fish, supported in most cases with collagen $\delta^{13}C_{coll}$ and $\delta^{15}N_{coll}$ values, from various localities. To evaluate the $\delta^{66}Zn$ deep-time preservation potential in enameloid, we also analysed fossil elasmobranch species from the Baltringer Formation of Germany, Globigerina Limestone of Malta, Na-arai Formation of Japan, Pungo River and Yorktown formations of North Carolina (USA) and Peace River and Tamiami formations of Florida (USA), covering the Early Miocene, Miocene-Pliocene transition, and the Early Pliocene (Supplementary Note 2). Details on the extant and fossil material used in this study (including museum catalogue numbers, locality, age, and stratigraphic context) are reported in Supplementary Data 1 with all $\delta^{66}Zn$, $\delta^{13}C_{coll}$, $\delta^{15}N_{coll}$ and $\delta^{18}O_P$ values and total body length estimates of individuals of fossil *Otodus chubutensis*, *O. megalodon*, *Carcharodon hastalis*, and *C. carcharias* based on tooth crown height, where applicable.

**Bioapatite zinc isotope analysis**. All teeth were cleaned by ultrasonication in ultrapure water (Milli-Q water) for 5 min and dried in a drying chamber at 50 °C. Enameloid samples were abraded from the top surface using a dental drill. Dentine samples were cut using a diamond-tipped cutting wheel. As teeth of extant *Raja clavata*, *Scyliorhinus canicula* and *Etmopterus spinax* were too small to separate enameloid from dentine, we measured 20 to 30 complete teeth of a single individual together and report the results here as "bulk teeth". For four fossil teeth, sediment adhering to the teeth was also sampled to evaluate the $\delta^{66}Zn$ values of the $CaCO_3$ sediment component of the final depositional environment of the teeth. All samples were dissolved in closed perfluoroalkoxy vials with 1 ml 1 M HCl on a hotplate for 1 h at 120 °C and then evaporated. The residue was then dissolved in 1 ml 1.5 M HBr and placed in an ultrasonic bath for 30 min. Zn purification was performed in two steps, following the modified ion exchange method adapted from Moynier et al.[49], first described in Jaouen et al.[9] and always included a chemistry

blank and reference standard (NIST SRM 1400) to monitor contamination and complete Zn elution. One ml of AG-1×8 resin (100–200 mesh) was placed in 10 ml hydrophobic interaction columns (Macro-Prep® Methyl HIC). The resin was then cleaned twice with 5 ml 3% $HNO_3$ followed by 5 ml ultrapure water. The resin was then conditioned with 3 ml 1.5 M HBr. After sample loading, 2 ml HBr were added for matrix residue elution, followed by Zn elution with 5 ml $HNO_3$. Following the second column step, the solution was evaporated for 13 h at 100 °C and the residue re-dissolved in 1 ml 3% $HNO_3$.

A total of 262 individual Zn isotope measurements were performed using a Thermo Fisher Neptune MC-ICP-MS at the Max Planck Institute for Evolutionary Anthropology (Leipzig, Germany). Instrumental mass fractionation was corrected by Cu doping following the protocol of Maréchal et al.[50] and Toutain et al.[51]. The in-house reference material Zn Alfa Aesar-MPI was used for standard bracketing. All $\delta^{66}Zn$ values are expressed relative to the JMC-Lyon standard material (mass dependent Alfa Aesar-MPI offset of +0.27‰ for $\delta^{66}Zn$[9,52]). Analysed sample solution Zn concentrations were close to 300 ppb, as was the Zn concentration used for the standard mixture solution. Zn concentrations in the respective samples were estimated following a protocol adapted from one used for Sr by Copeland et al.[53], applying a regression equation based on the Zn signal intensity (V) of three solutions with known Zn concentrations (150, 300, and 600 ppb). The $\delta^{66}Zn$ measurement uncertainties were estimated from standard replicate analyses and ranged between ± 0.01‰ and ± 0.04‰ (2 SD). Samples were typically measured at least twice with mean analytical repeatability of <0.02‰ (2 SD, $n = 235$). Reference material NIST SRM 1400 was prepared and analysed alongside the samples and had $\delta^{66}Zn$ values (+0.92 ± 0.03‰, $n = 28$) as reported elsewhere[13,54] demonstrating complete Zn elution during column chromatography. Reference materials and samples show a typical Zn mass-dependent isotopic fractionation, i.e., the absence of isobaric interferences, as the $\delta^{66}Zn$ vs. $\delta^{67}Zn$ and $\delta^{66}Zn$ vs. $\delta^{68}Zn$ values fall onto lines with slopes close to the theoretic mass fractionation values of 1.5 and 2, respectively (Supplementary Data 1).

**Organic carbon and nitrogen isotope analysis**. Collagen was isolated to determine the stable isotope composition of organic carbon and nitrogen from extant elasmobranch and teleost tooth dentine and teleost bone. Due to the small tooth size of *Etmopterus spinax*, we measured jaw cartilage and complete bulk teeth from this individual (Supplementary Data 1). Powdered dentine samples were collected from all modern teeth and bones using a low-speed handheld drill with a 300-micron diamond-tipped bit. These samples were then demineralised using chilled 0.1 M HCl (modified method after Brown et al.[55]). After demineralisation, samples were rinsed five times with deionized water and freeze-dried overnight. Samples were weighed out to 0.4–0.5 mg in 3×5 mm tin capsules. All collagen samples were measured for $\delta^{13}C$ and $\delta^{15}N$ values using a Costech 4010 Elemental Analyzer coupled to a Delta V Plus continuous-flow isotope ratio mass spectrometer with a Conflo IV in the Stable Isotope Ecosystem Lab at the University of California Merced. All data were corrected for linearity and drift using a suite of calibrated reference materials (USGS 40 [$n = 16$, $\delta^{15}N = -4.5 ± 0.3$‰, $\delta^{13}C = -26.4 ± 0.1$‰]; USGS 41a [$n = 9$, $\delta^{15}N = 47.6 ± 0.2$‰, $\delta^{13}C = 36.6 ± 0.1$‰]; costech acetanilide [$n = 6$, $\delta^{15}N = -0.8 ± 0.2$‰, $\delta^{13}C = -28.3 ± 0.1$‰]). Long-term instrumental standard deviation was determined to be 0.1‰ for $\delta^{13}C$ and 0.3‰ for $\delta^{15}N$.

**Bioapatite phosphate oxygen isotope analysis**. Bioapatite phosphate oxygen isotope ($\delta^{18}O_P$) analysis was performed on a subset of fossil teeth from Germany and Malta (Supplementary Fig. 8, Supplementary Data 1). Enameloid powders were converted to silver phosphate ($Ag_3PO_4$) for oxygen isotope measurements of bioapatite phosphate by digestion with hydrofluoric acid (HF) and crash precipitation of $Ag_3PO_4$ following an adapted version of the rapid precipitation protocol developed by Dettman et al.[56] and modified by Tütken et al.[57] (see a detailed description in Supplementary Methods).

Oxygen isotope delta measurements of $Ag_3PO_4$ were conducted using a high-temperature elemental analyser (TC/EA) coupled to a Delta V isotope ratio mass spectrometer via a Conflo IV interface (Thermo Fisher Scientific, Bremen, Germany) at the Max-Planck-Institute for Evolutionary Anthropology in Leipzig (MPI-EVA; see Supplementary Methods for instrumentation and analysis details). Samples were introduced using a Costech Zero Blank Autosampler (Costech International, Cernusco sul Naviglio, Italy) and converted to CO using a reactor temperature of 1450 °C. Gases were separated using a Eurovector E11521 1.4 m x 4 mm×6 mm stainless steel GC column with 80/100 mesh 5 Å molecular sieve packing (Eurovector Instruments & Software, Pavia, Italy) maintained at 120 °C with a carrier gas pressure of 1.3 bar.

Scale calibration to VSMOW-SLAP scale was conducted using two-point scale normalisation using commercially available standards or in-house standards that were in turn calibrated to international reference materials. Calibration and quality control methods are the same as outlined in Pederzani et al.[58]. For two-point scale normalisation we used the B2207 silver phosphate standard ($\delta^{18}O = 21.7 ± 0.3$‰, 1 SD; Elemental Microanalysis, Okehampton, UK) and an in-house silver phosphate standard (KDHP.N, $\delta^{18}O = 4.2 ± 0.3$‰, 1 SD). The accepted value of KDHP.N was determined in turn by two-point calibration using B2207 and the international reference material IAEA-SO-6 (barium sulphate, $\delta^{18}O = -11.4 ± 0.3$‰, 1 SD)[59]. To check isotopic consistency across silver phosphate precipitations and to ensure equal treatment, aliquots of an in-house

modern cow enamel standard (BRWE.2) and the standard material NIST SRM 120c (formerly NBS 120c) were precipitated and measured alongside each batch of samples. As a quality control to check the consistency of the scale calibration across runs independent of silver phosphate precipitations, we additionally used a commercially available silver phosphate (AS337382, Sigma Aldrich, Munich, Germany). Measurements of these quality control standards gave $\delta^{18}O$ values of $14.6 \pm 0.7\permil$ for BRWE.2 (1 SD, $n = 10$), $21.3 \pm 0.6\permil$ for NIST SRM 120c (1 SD, $n = 12$) and $14.1 \pm 0.2\permil$ for AS337382 (1 SD, $n = 32$). These results compare well to consensus values for NIST SRM 120c of $21.7\permil$[60], as well as long-term averages for BRWE.2 of $14.5 \pm 0.4\permil$ and AS337382 of $14.0 \pm 0.3\permil$. Samples were typically measured in triplicate, and the average reproducibility of sample replicate measurements was $0.3\permil$ (see Supplementary Methods).

**Electron microprobe analysis**. To compare Zn distribution patterns among equivalent extant and fossil teeth, thin sections were made from a *Galeocerdo aduncus* tooth (Early Miocene, Germany) and an extant *G. cuvier* from KZN. X-ray element distribution maps were acquired using a field-emission electron microprobe "SX5FE" from Cameca. We used an acceleration voltage of 20 kV. As zinc is only contained in trace amounts, we had chosen a high probe current of 200 nA (in some cases even 500 nA). Furthermore, the intensity of the Zn Kα line was measured simultaneously on three wavelength-dispersive spectrometers. Depending on the size of the mapped area, the dwell-time and step width were varied (D.Time = 45–60 ms per pixel; step Width = 1–6 μm).

**Statistical analysis**. Analysis of variance (ANOVA) was performed to determine statistical differences in $\delta^{66}Zn$ values across fossil assemblages, among fossil species and to compare fossil to extant species. To adhere to ANOVA's assumptions, the $\delta^{66}Zn$ datasets of assemblages, periods or species that were compared underwent visual inspection to check for normally distributed and homogeneous residuals, as well as equal variance using Levene's test. When the ANOVAs identified significant difference ($p$ value < 0.05) among assemblages, periods or species, post-hoc Tukey pairwise comparisons were carried out to determine which populations were significantly different ($p$ value < 0.05) from others in terms of their $\delta^{66}Zn$ values. In cases where an unequal variance or unequal sample size was found between groups, Games-Howell pairwise-comparison post-hoc test was performed instead. When applicable, $p$ values were adjusted using the Holm-Bonferroni or Tukey correction for multi-testing.

Statistically significant differences in $\delta^{66}Zn$ values between species was determined through ANOVA for given assemblages (Germany (Miocene, $n = 59$): $F(7,51) = 9.35$, $p = <0.0001$; Malta (Miocene, $n = 30$): $F(5,24) = 5.19$, $p = 0.002$; North Carolina (Miocene, $n = 13$): $F(3,9) = 4.41$, $p = 0.036$; and North Carolina (Pliocene, $n = 28$): $F(3,24) = 9.11$, $p = 0.0003$). Other assemblages were not investigated as their sample size were deemed too small ($n \leq 7$). ANOVA were also conducted on given genera that are represented in most assemblages and time periods, namely *Carcharhinus* spp. ($n = 22$; $F(3,18) = 6.97$, $p = 0.003$), *Carcharias* spp. ($n = 26$; $F(5,17) = 1.7$, $p = 0.19$), *Carcharodon* spp. ($n = 31$; $F(4,26) = 3.7$, $p = 0.016$), *Galeocerdo* spp. ($n = 22$; $F(3,18) = 0.98$, $p = 0.42$) and *Otodus* spp. ($n = 42$; $F(5,36) = 5.2$, $p = 0.001$). Finally, ANOVA were also conducted between *Carcharodon* spp. and *Otodus* spp. ($n = 73$; $F(4,68) = 9.81$, $p = <0.001$). Here, all regions and periods were grouped solely based on the species, with the exception of the extant *Carcharodon carcharias* that was kept separately (Fig. 4). The test was performed again, where *Otodus megalodon* specimens from Japan (Pacific) were excluded from the analysis, on the basis of focussing on the Paratethys/Tethys and Atlantic oceans ($n = 68$; $F(4,63) = 10.56$, $p = <0.001$).

All statistical analyses were conducted using the open-source program R software[61] (R version 4.0.2) using an alpha level for significance of 0.05. The results of ANOVAs, and post-hoc Tukey and Games-Howell pairwise comparisons can be found in Fig. 4 and Supplementary Table 4–12.

**Total length estimations of *Otodus chubutensis*, *O. megalodon*, *Carcharodon hastalis*, and *C. carcharias***. For this study, we used Shimada's[62] linear functions showing the relationship between the tooth crown height (CH; maximum vertical enameloid height) and total body length (TL) in extant *Carcharodon carcharias* to estimate the TL of individuals of *Otodus chubutensis*, *O. megalodon*, *C. hastalis*, and *C. carcharias* from the CH of examined fossil teeth. Tooth crown height was measured (or estimated where possible if broken) to the nearest millimetres for each tooth. Then, the tooth position in the jaw for each fossil tooth was determined morphologically using illustrations following Uyeno & Matsushima[63], Uyeno et al.[64], and Ehret et al.[65] for *Carcharodon* spp., and Perez et al.[66] for *Otodus* spp. The reliability of a TL estimation is known to deteriorate towards the distal end of each dentition progressively, and the TL predictability is less reliable for lower teeth than upper teeth[16,62,66]. Thus, the TL for fossil teeth that were determined to have been located distal to the fourth lateral tooth ('L4' of Shimada[62]) in the upper dentition, and all teeth distal to the second anterior tooth ('a2' of Shimada[62]) in the lower dentitions, were not calculated. It should be noted that Perez et al.'s[66] TL estimation method for *Otodus* species that require crown width (CW) measurements are not readily applicable to our many incomplete samples where CH is more feasible to infer with confidence than CW especially for specimens with both

mesial and distal ends missing. The CH measurement assumed original tooth position and estimated TL value for each tooth are given in Supplementary Data 1.

**Reporting Summary**. Further information on research design is available in the Nature Research Reporting Summary linked to this article.

## Data availability

All $\delta^{66}Zn$, $\delta^{13}C_{coll}$, $\delta^{15}N_{coll}$ and $\delta^{18}O_P$ values and total body length estimates of individuals of fossil *Otodus chubutensis*, *O. megalodon*, *Carcharodon hastalis*, and *C. carcharias* based on tooth crown height generated in this study are provided in the Supplementary Information, Supplementary Data 1, and Source Data file. All museum catalogue numbers are provided in Supplementary Data 1.

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

## Acknowledgements

The Max Planck Society supported this research. We thank Elmar Unger for providing sample material. Sven Steinbrenner and Manuel Trost (Department of Human Evolution, Max Planck Institute for Evolutionary Anthropology, Leipzig) are thanked for technical support. This research was funded in part by a National Science Foundation Sedimentary Geology and Paleobiology Award to M.L.G. and M.A.B. (Award Number 1830581), K.S. (Award Number 1830858), S.L.K. (Award Number 1830480), R.A.E. (Award Number 1830638), the Deutsche Forschungsgemeinschaft ('PALÄODIET' project no. 378496604) and T.T. and K.J. received funding from the European Research Council under the European Union's Horizon 2020 research and innovation program (grant agreement no. 681450 and no. 803676). We also thank John R. Nance and Stephen J. Godfrey (Calvert Marine Museum, Solomons, Maryland, USA) and Kirsten Grimm (Osteological Collection, Institute of Geosciences, Mainz, Germany) for the curation of and access to sample material.

## Author contributions

J.M., M.L.G. and T.T. conceived the study. J.M., M.L.G., K.S., S.L.K., H.M., M.A.B., G.S.-V., N.S., J.P., R.A.E. and T.T. selected the sample material. J.M. performed the zinc isotope analyses. S.L.K. and M.K. performed the carbon and nitrogen isotope analyses. S.P. performed the oxygen isotope analyses. N.J. performed the electron microprobe analyses. N.B. performed the statistical analyses. K.S. performed the total length estimations. J.-J.H., S.L.K., K.J., N.J. contributed reagents/analytic tools. J.M. wrote the original draft. J.M., M.L.G., K.S., S.L.K., H.M., M.A.B., N.B., K.J., N.S., J.P. and T.T., analysed the data, wrote, and edited the manuscript.

## Funding

## Competing interests

The authors declare no competing interests.
