## [Peer Review File · Nature Communications]

Trophic position of *Otodus megalodon* and great white sharks through time revealed by zinc isotopesReviewers' Comments:

Reviewer #1:

Remarks to the Author:

Review for NCOMMS-21-38599

Overview: In this manuscript, the authors study the hypothesis that a dietary competition is responsible for the extinction of the megatooth shark (*Otodus megalodon*), thus putting aside climatic factors for instance. They used zinc isotopes to assess the trophic level of this shark species and compare with other shark species (extant and extinct) to estimate the possible niche overlap in diet preferences. According to the authors, they found significant difference in zinc isotope ratios among *Otodus* populations, which is interpreted as a change in prey over time. The authors also claimed that the great white shark shifted its diet in the Pliocene, and that could support the hypothesis of competition between the megatooth and great white shark.

General statement

Overall, I think this is a fine study using non-traditional stable isotopes that deserves to be published, but not in Nature Communications. I appreciated the effort to analyze different tissues (even if the sample sizes were really tiny for tissues other than enameloid) and the mention of different factors that could influence $\delta^{66}\text{Zn}$ variation. The methods look fine but I think that the authors are overselling their results (and the title). The main goal of the paper is to test if resource competition could have driven *O. megalodon*'s extinction. Their conclusion is that *Carcharodon* sharks outcompeted *O. megalodon*, the latter decreasing its trophic position in the Pliocene relative to the Miocene. This, however, is only observed in the samples from North Carolina. Early Pliocene *Otodus* from Japan shows values as low as those observed in other places during the early Miocene. I have a number of concerns that preclude the publication in Nature Communications (see below).

Major comments

A major issue is comparing species from different places because variation in Zn baseline values is something that should deserve more concern. I don't know much about Zn isotopes but there are different forms of inorganic zinc in the ocean (Zn^{+2} , $\text{Zn}(\text{OH})_2$, ZnHCO_3^- , ZnCO_3) that could influence isotopic fractionations at the base of the food web. Ideally, this idea of competition and trophic changes could be tested in Miocene and Pliocene fossil sharks from the same place but as a paleontologist I can understand the limitations of the fossil record. This reminds me a little bit of the origins of $\delta^{15}\text{N}$ studies, where baseline variations of nitrogen were basically disregarded and samples from different localities were directly compared. Many studies and decades had to pass for us to acknowledge the huge variations in the baseline values and understand the limitations of the method. In any case, this study should spur further research on the topic.

Based on the aforementioned comment, another main concern is about the goal of the paper and the "Implications for the megatooth shark extinction" (starting line 267). The interpretation is finally very speculative, and the strong statement "the $\delta^{66}\text{Zn}$ values in this study lend support for *Carcharodon* driving the extinction of *Otodus* through resource competition" (lines 275-275) is very hard to buy. I don't see the evidence for direct competition between the *Otodus* and *Carcharias* lineages. I rather see niche partitioning.

I think the authors have made a focus on shark interactions, but it is overlooked that species interactions could have acted between unrelated clades, such as cetaceans. The first taxon that is ignored in this study is the killer whale (*Orcinus orca*). It could be also linked to the extinction of the megatooth shark. Given the fossil record of *Orcinus*, it appears that *Orcinus orca* is known from Japan (Hokkaido), in the Pliocene (Zanclean: 5.332-3.6 Ma). After that *O. orca* spread around the world since the late Pliocene. So, this taxon could have also played a role in the demise of the megalodon.

Another obvious unnoticed taxon could be the interaction with the extinct *Leviathan melvillei*. Indeed, the two giant marine predators, *O. megalodon* and *L. melvillei*, were found in the same mysticete-rich Serravallian locality (Lambert et al. 2010 – Nature) and studying their interaction might be relevant to understand the extinction of megalodon. Furthermore, large physeteroid teeth found in other Miocene localities worldwide indicate that giant raptorial sperm whales occupied a top predator position in various marine regions during the Miocene, a role now mostly taken by *Orcinus*. As such, the appearance of these sperm whales probably had a profound impact on the structuring of middle to late Miocene marine communities and food chains, like the impact of *Orcinus*.

You would certainly argue that *L. melvillei* cannot be responsible for the extinction of the megatooth shark because of time difference between the two lineages (*L. melvillei* went extinct earlier), but you cannot reject the hypothesis that their possible interactions could have mediated the diet shift of the megalodon. Same rationale with *Orcinus*, which more actively predate mysticete while the great white shark is more a scavenger of dead whales.

This is to say that the view presented here is very simplistic and incomplete. Therefore, the conclusions presented, including the title, are overstated. Their results rather point to niche partitioning rather than direct interactions that could have led to extinction of a lineage.

Some results are hard to believe, especially the fact that the mean $\delta^{66}\text{Zn}$ value for extant *C. carcharias* places the great white shark at a higher trophic level than the Atlantic *O. megalodon*. The authors acknowledge the limited sample size but then speculate on diet preferences of Pliocene great white and megatooth sharks.

An important neglected aspect is that fossil occurrences inform on first and last appearance datum (FAD and LAD) of a taxon but given the various biases in the fossil record these FAD and LAD are not the 'true' ages that determine the species' lifespan. Based on an analysis of the lamniform fossil record (Condamine et al. 2019 - PNAS), the origin of *C. carcharias* is estimated at 6.76 Ma (95% CI: 5.45-8.59 Ma), while the origin of *O. megalodon* is estimated at 23.3 Ma (95% CI: 22.02-24.90 Ma). Importantly, the extinction of *O. megalodon* is estimated at 0.754 Ma (95% CI: 0.003-1.96 Ma), a bit more recent than the LAD in the Zanclean. This suggests that the interactions between *C. carcharias* and *O. megalodon* could have lasted longer and that niche partitioning was an efficient way to avoid direct competition. This very same study showed diversity dependence within the large-sized lamniform species (including *C. carcharias* and *O. megalodon*) so that rates of species formation and extinction depend on the number of species of the same guild.

Why not using calcium (Ca) isotope values from tooth enamel (expressed as $\delta^{44}/^{42}\text{Ca}$) to investigate resource partitioning? This has been done in theropod dinosaurs for instance (Hassler et al. 2018 – PRSB). Because Ca isotope ratios are widespread, it is possible to infer the ancestral diets (for instance, fishes for spinosaurs). This provides more clues than zinc isotopes, it seems.

Finally, I would include a bivariate plot of $\text{d}18\text{O}$ and $\text{d}66\text{Zn}$ because the authors use results from both isotopes to infer potential preys (Zn for trophic position and oxygen to infer where they could have preyed within the water column)

Minor comments

Lines 59-60: "likely provides better long-term preservation than organic collagen-bound nitrogen¹¹, the most commonly used isotopic trophic tracer to date". What about calcium isotopes that have been used quite a lot, in dinosaurs for instance?

Lines 63-66: I found the presentation of the megalodon quite short. We know much more than what is said here. I don't agree with the idea that its extinction remains an enigma.

- Figure 1: Why are the South African specimens separated from the rest? I guess what is really not clear to me is the grouping of the second box that includes species from the US, Island, Norway, etc.
- Figure 2: I would find the figure more visually informative if the species were arranged in decreasing or increasing order of their values.
- Figure 3: What are the dark squares in *Otodus*? That's not explained in the legend

Reviewer #2:

Remarks to the Author:

This paper reports a study to examine trophic level distinctions amongst extinct shark taxa from the Miocene to the Pliocene based on the determination of Zinc isotopes ($^{66}\text{Zn}/^{64}\text{Zn}$) in shark tooth enameloid, having first examined these and other isotopic (carbon nitrogen, oxygen) compositions in extant shark's teeth. Zn isotopes are a rather new means for examining trophism, that to date have been applied in a very limited way in marine contexts. Shark teeth lend themselves to this application because their teeth are diagnostic to species, they are well-preserved over long periods of time, their structures are well-understood, and their size reflects in some degree the size of the animal. The ultimate focus/hypothesis in this study is on distinctions in trophic level between the giant extinct megatooth shark (*Otodus megalodon*) and the (still extant) Great White shark (*Carcharodon carcharias*) that might have led to competition and eventual extinction of the former.

The authors first establish the 'credentials' of Zn isotopes by examining and comparing multiple isotopic and trace element systems of several extant shark taxa from different ocean regions. Zn isotopes are compared against carbon and nitrogen isotopes from associated organic components, showing that nitrogen isotopes, the "standard" trophic indicator in marine system, compare well, but it is more variable than Zn isotopes. From this it is deduced that Zn isotopes may provide a more globally standard indicator, although certainly this remains to be tested further. Tests of trace element distribution to assess diagenetic invasion of the fossil tooth enameloid are carried out, which identified areas to avoid. The stated similarity of phosphate oxygen isotopes in fossil versus extant teeth was not very convincing to this reviewer, from the little information we are provided about the palaeoceanographic contexts. It should be relatively simple to back up this similarity in present vs past ocean temperatures with hard data from the palaeoceanographic marine core literature. Still, I think that overall, the case is convincing. From phosphate any sort of influence on the results. Zn and other isotope analyses are then carried out on a range of extant taxa of various sizes, typical prey, and from different oceanic regions, and statistically compared. Given that this is a very complex dataset the rationale for using largely pairwise statistical comparisons for each isotope and species was not clear (nor were the results clear). It would have been preferable to present the data using the tools commonly used for isotopic niche distinctions amongst ecologists, for instance Standard ellipse areas (SEAs) for binary isotope systems, or even NicheROVER (Swanson et al. 2015) for multi-dimensional systems. These allow both statistical comparisons as well as far better visualisation of the data.

There are two other areas of concern. First, it is not at all clear that the main hypothesis, that the giant extinct megatooth shark (*Otodus megalodon*) was outcompeted by the Great White shark (*Carcharodon carcharias*) can be tested using this data, and given gaps in the information provided. The Zn isotope data do show some differences between the two taxa but these differences vary according to both time and oceanic region. The hypothesis is too broad, and other possibilities are not entertained. Fig 2 for instance shows significant differences in Zn isotope composition for *O. megalodon* in the early Pliocene of Japan (lower values = higher trophic level) vs same period for North Carolina. It's not clear how the conclusion is reached that

Carcharodon carcharias outcompeted *Otodus megalodon* based on this data. There might be, for instance, regional oceanic conditions (currents?) and regional prey species that differ. Overall, oceanographic contextualisation is very sparse in this study. This is a problem for several reasons, including some recognition of ocean currents and temperatures, in relation to productivity and the potential array of prey species. Because the oceanography is not clear and not referenced or explained, the conclusion about similar temperatures derived from PO₄ oxygen isotopes in past and present could be challenged. There is also the question of the influences of continental dust on ocean systems, depending on where in the ocean. We know that continental dust is far-reaching and influences ocean productivity (for instance Saharan dust across the Atlantic), but further, several studies have shown that Zn isotopes can be used as tracers for the dust clouds since Zn varies isotopically by source. In other words, there are other possible ways in which Zn isotopes might vary by region. Such issues need addressing if the conclusions drawn on the basis of comparisons from different periods and places can be accepted with confidence.

The next issue is of a different sort. This paper seems (to this reviewer) to consist of two natural sections, each of which has a large amount of complexity and test, but putting them together makes it rather difficult to see the wood for the trees, so to speak. The section on extant species and how Zn isotopes behave for different taxa, conditions and prey, and in relation to the other main trophic indicator, nitrogen isotopes, is of course essential fundamental information to tackle the fossil issues, but on its own it still has some important contributions for modern shark ecology or more broadly, marine trophic systems (the nitrogen vs Zn isotope differences for instance). But the complexity of the many comparisons amongst numerous taxa in different places and conditions, and their tests for significance, make it very hard for a reader to distinguish the important features. The authors might consider reporting first their modern data? In doing so they also use some of the tools developed by ecologists for niche breadth comparisons within ecosystems such as Standard Ellipse Areas (SEAs) for binary isotope systems or better still the new NicheROVER (Swanson et al. 2015) for multi-dimensional systems. That would be really useful to both the ecological and palaeontological communities. A stronger palaeoceanographic context would provide a more solid basis for interpretation of the fossil data .

Minor comments.

Repetition of the word "iconic" was somewhat jarring.

Statistical analysis (Line 400 onwards)

Analysis of variance (ANOVA) was performed to determine statistical differences in $\delta^{66}\text{Zn}$ values across fossil assemblages or among fossil and extant species but the text only mentions the fossils? Is this just a sentence error?

the first sentence of the abstract "The demise of the iconic gigantic Neogene megatooth shark, *Otodus megalodon*, remains enigmatic, ..." does to make complete sense. You mean the "reasons for the demise"?

also in the abstract, the statement "Our $\delta^{66}\text{Zn}$ values are consistent with the hypothesis that *C. carcharias* outcompeted *O. megalodon*." is not supported by the data provided, would need more careful wording.

Line 56: "Studies to date indicate that $\delta^{66}\text{Zn}$ values successively decrease with increasing trophic level 9-13". Its worth pointing out that v few of them have been applied in marine, and then mostly in quite circumscribed archaeological contexts.

Line 77. "by measuring tooth phosphate oxygen isotopes ($\delta^{18}\text{O}$)". Was this done on all the extant and fossil teeth, not clear though I may have missed?

Line 97. "We also observe a correlation between trophic level and $\delta^{15}\text{N}$ values ($R^2 = 0.42$, $p = 6.97\text{E-}6$, $n = 40$)". Bit surprised that its quite a weak correlation, is it because of the odd pesky outlier? Quite useful to id the outliers esp. since you conclude (rightly) that nitrogen isotopes are more dependent on local conditions.

Line 138/139. This statement "We observe no correlation between the total body length of *Otodus* spp., *Carcharodon hastalis* and *C. carcharias* and their respective $\delta^{66}\text{Zn}$ values.." is also surprising especially since multiple marine studies show that marine animals tend to consume larger (and hence

higher trophic level) prey as body size increases. Do the uncertainties in estimating body length interference with this analysis?

Line 175. "Zinc isotope variability among marine organisms and their tissues is largely unknown, currently limiting our ability to identify specific food items based on shark enameloid $\delta^{66}\text{Zn}$ values beyond generally observed trophic level effects" . Might this be another basis for being more cautious in the conclusion that *O. megalodon* was outcompeted by *C. carcharias*?

Line 243. "For the *Carcharhinus* teeth from Malta, this interpretation is supported by lower $\delta^{18}\text{O}$ values than sympatric species, indicating a higher water temperature or lower salinity: i.e., a shallow and/or brackish water habitat (Supplementary Figure 11)." Where in time in relation to the Messinian crisis?

Line 283. "However, during the early Pliocene, the *Otodus* lineage represented by *O. megalodon* shows a considerable increase in the mean $\delta^{66}\text{Zn}$ value for the Atlantic populations, hinting at a reduced trophic position for the megatooth shark lineage in the Atlantic." Where about in the Atlantic? See comments on Saharan dust, know to be active from Miocene onwards.

Line 293. "The extinction of *Otodus megalodon* could still be caused by multiple factors, including climate change³⁴. However, the apparent synchronous global occurrence of *Carcharodon carcharias* with the extinction of *O. megalodon*¹⁵". This is rather posthoc and not convincing.

Line 351. "bone"? first mention of bone, this is a very different element and in sharks?

Supplementary I.

Line 45. "Heavy Zn isotopes tend to be partitioned into stiffer bonds " . Rather inelegant, can be improved

Line 65 onwards. "The isotopic composition of marine dissolved Zn below 500 m seems to be globally homogenous with values close to +0.5 ‰, despite variable Zn concentrations^{14,15}. Consequently, the bulk isotopic composition of dissolved marine Zn is enriched in ^{66}Zn relative to its major inputs from rivers and aeolian dust, which centre on the global crustal average of +0.3 ‰^{16,17}." This is rather ignored in the main paper where very little information about where samples found and the surrounding ocean conditions.

Line 284. "Another possible explanation for the $\delta^{66}\text{Zn}$ differences observed between the KZN shark species could be taxon-dependent diet-bioapatite Zn discrimination factors." Far-fetched possibility! rather consider the conditions in that area and prey species.

SI Fig 2. What does "bulk teeth" mean? Did I miss an explanation somewhere?

Reviewer #3:

Remarks to the Author:

This new study by McCormack et al. tests the hypothesis that competition between the extinct megatoothed shark *Otodus megalodon* was outcompeted by the modern great white shark *Carcharodon carcharias* during the Pliocene through the analysis of Zinc isotope ratios from a wide variety of modern and fossil shark and fish teeth. Zinc isotopes have not been previously studied in nonmammals, and this method is a creative and novel approach to reconstructing ancient food webs. Critically, this study has produced a surprisingly large volume of zinc isotope data to establish a baseline for modern species (captive and wild), and contemporary as well as stratigraphically separated shark specimens. I am not a geochemist, but nevertheless find the statistical results convincing. I am quite impressed with this study, and only have minor comments. Sincerely, Robert W. Boessenecker, Ph.D., College of Charleston, South Carolina, USA

1) Are any of the fossil specimens phosphatized (e.g. blackened/mineralized and/or dense)? Shark teeth from the Lee Creek Mine and Bone Valley Formation (sources of some of the analyzed specimens) are often phosphatized. Based on discussions about uptake in heavy metals, the early diagenetic mineralization of shark teeth might similarly affect zinc isotope ratios.

2) The Pliocene-Pleistocene marine megafaunal extinction has not been dated, and post-dates the extinction of *O. megalodon*. The study by Pimiento et al. (2017) is a great one, but their faunal

analysis used Pliocene and Pleistocene time bins – necessary given the coarseness of the available data – and the contrast in taxa between these two time bins artificially assigns the Pliocene-Pleistocene boundary as the time of the extinction. However, it could be late Pliocene or early Pleistocene (~800 ka to 2.5 Ma), and it is possible that the extinction is asynchronous in different ocean basins. It's also possible that it was a period of more rapid faunal turnover rather than a classic example of an extinction event. See Boessenecker (2013: *Geodiversitas*: 35:4:815-940) for more on marine mammal extinctions/faunal turnover. Regardless, thanks to late Pliocene marine vertebrate assemblages, it's clear that the extinction of *O. megalodon* pre-dates this event by at least a million years or so (Boessenecker et al., 2019).

3) It is critical to note that while interesting, the studies on bite marks are qualitative (e.g. "we found some bite marks on this whale bone, how neat") and a quantitative study of bite marks currently does not exist. Ideally, some study out there would compare presence/absence of bite marks for hundreds of specimens from stratigraphically separated assemblages and compare changes in trace sizes and prey taxa composition through time.

241-3: this suggests that *Galeocerdo* spp. trophic level did not change despite an increase in body size – perhaps this is worth an extra comment.

297-8: I'm not sure if this helps, but I wonder how much of this has to do with middle Miocene mysticetes being relatively less abundant than Pliocene mysticetes (e.g. mid Miocene assemblages are dominated by odontocetes with rare mysticetes).

300: As written, this citation of Pimiento and Balk suggests that they proposed climate change as a driver of *O. megalodon* extinction: they tested this, did not find evidence supporting it, and highlighted instead a biotic driver. This critical analysis is what led us to identify *Carcharodon carcharias* as the most likely biotic driver (which led you all to undertake this study). I suggest more carefully constructing the sentence before this citation to make it clear that these authors excluded climate change as a likely driver.

Reviewer #4:

Remarks to the Author:

The manuscript include important information about the use of Zinc isotope to found the trophic changes in Neogene megatooth shark, *Otodus megalodon*, a fossil shark tooth enameloid. Also they take samples from recent sharks and fish to do their comparison with fossil sharks.

I have some questions about the manuscript:

L146: In the manuscript you mentiona about the fish used in your results: *Sparus auratus* from pisciculture. This species feed on controlled kind of food, comparing with the species in the wild. Why to use a fish and not a shark in aquarium, which feed on specific diet?

L 166 What are the main prey for mako shark and white shark in the wild? to see differences on zinc isotopes by each species.

L168-169 Would you explain more about the differences of zinc isotopoe on marine ecosystem and terrestrial ecosystem?

L171 In the manuscript you mention about *Galeocerdo cuvier* as an offshore pelagic shark; however this shark species are more from coastal habitats feeding on turtles, seabird, etc

Author response

We are thankful for the very thorough and constructive feedback given by all reviewers. Implementing the reviewers' comments has significantly strengthened our arguments. Specifically, we have toned down our conclusions regarding the competition between *O. megalodon* and *C. carcharias* following the suggestions of reviewer 1 and 2 and the editor. In doing so, we also changed the title of the manuscript.

In addition, and complying with the editorial requests, we have combined the results and discussion sections. According to the Nature Communications formatting instructions, headings are not allowed in the discussion but required for the results. Still, multiple Nature Communications papers have combined both results and discussion (e.g., Santora et al., *Nat Commun* 12, 6492 (2021)) which we believe is also most appropriate in our case.

In the following we discuss the comments made by each referee (original remarks are in black, our response in blue). We also numbered the reviewer comments, allowing us to refer to comments made by each reviewer and our responses. When referring to changes made with line numbers, we refer to the tracked changes versions of the manuscript and Supplementary Information unless otherwise specified.

Reviewer #1 (Remarks to the Author):

Review for NCOMMS-21-38599

Overview: In this manuscript, the authors study the hypothesis that a dietary competition is responsible for the extinction of the megatooth shark (*Otodus megalodon*), thus putting aside climatic factors for instance. They used zinc isotopes to assess the trophic level of this shark species and compare with other shark species (extant and extinct) to estimate the possible niche overlap in diet preferences. According to the authors, they found significant difference in zinc isotope ratios among *Otodus* populations, which is interpreted as a change in prey over time. The authors also claimed that the great white shark shifted its diet in the Pliocene, and that could support the hypothesis of competition between the megatooth and great white shark.

General statement

1.1

Overall, I think this is a fine study using non-traditional stable isotopes that deserves to be published, but not in Nature Communications. I appreciated the effort to analyze different tissues (even if the sample sizes were really tiny for tissues other than enameloid) and the mention of different factors that could influence $\delta^{66}\text{Zn}$ variation. The methods look fine but I think that the authors are overselling their results (and the title). The main goal of the paper is to test if resource competition could have driven *O. megalodon*'s extinction. Their conclusion is that *Carcharodon* sharks outcompeted *O. megalodon*, the latter decreasing its trophic position in the Pliocene relative to the Miocene. This, however, is only observed in the samples from North Carolina. Early Pliocene *Otodus* from Japan shows values as low as

those observed in other places during the early Miocene. I have a number of concerns that preclude the publication in Nature Communications (see below).

We are glad that the reviewer understands the scientific significance of our study. We addressed the reviewer's concerns and modified the manuscript where appropriate; most notably, we have toned down our conclusions regarding the competition between *O. megalodon* and *C. carcharias* and, as a result, changed the title to "Trophic position of *Otodus megalodon* and great white sharks through time revealed by zinc isotopes".

We hope the reviewer will now be able to further appreciate the novelty of our study and the multi-disciplinary implications of our results. Namely, we have introduced an entirely new and powerful marine dietary and trophic level proxy for marine vertebrates with important implications for studies on modern marine ecosystems, research on ecosystem response to climate change, and species conservation issues, among many more. Moreover, the introduction of a reliable deep-time trophic level proxy also has important implications for future palaeontological research, as illustrated in our manuscript, which presents the most comprehensive geochemical trophic ecology study of Neogene sharks (or any vertebrate group to our knowledge) to date. We have also clearly demonstrated differences, and, in some cases, similarities in the diet for taxonomic groups in time with significant consequences for their respective evolution. Therefore, we believe the multi-disciplinary nature and novelty of our research merits its publication in Nature Communications.

Major comments

1.2

A major issue is comparing species from different places because variation in Zn baseline values is something that should deserve more concern. I don't know much about Zn isotopes but there are different forms of inorganic zinc in the ocean (Zn^{+2} , $Zn(OH)_2$, $ZnHCO_3^-$, $ZnCO_3$) that could influence isotopic fractionations at the base of the food web. Ideally, this idea of competition and trophic changes could be tested in Miocene and Pliocene fossil sharks from the same place but as a paleontologist I can understand the limitations of the fossil record. This reminds me a little bit of the origins of $\delta^{15}N$ studies, where baseline variations of nitrogen were basically disregarded and samples from different localities were directly compared. Many studies and decades had to pass for us to acknowledge the huge variations in the baseline values and understand the limitations of the method. In any case, this study should spur further research on the topic.

We thank the reviewer for this comment. We agree that marine $\delta^{66}Zn$ food web baseline variations are poorly understood and that it is important to properly assess them. That is why we included a state-of-the-art section in the Supplementary Information regarding marine $\delta^{66}Zn$ values and the absence of studies regarding baseline values for $\delta^{66}Zn$ (lines 65 to 77 original Supplementary Information). We also discuss baseline ambiguities as a potential factor limiting our interpretations (e.g., lines 216 to 218, 259 to 266, original manuscript). Our $\delta^{66}Zn$ -based approach is so novel that there are no studies on marine Zn isotope baselines yet. So far, only two other studies (Jaouen et al., 2016; McCormack et al., 2021) have

even examined Zn isotopes in marine vertebrates (limited to mammals) as an ecological proxy before. Nevertheless, our study is significant because it provides an extensive $\delta^{66}\text{Zn}$ dataset of both modern and fossil shark teeth values, which demonstrate that $\delta^{66}\text{Zn}$ values within a species or a taxonomic group (with similar trophic levels) are generally directly comparable in their values independent of geologic age and geographic location. This is a remarkable observation because it indicates that $\delta^{66}\text{Zn}$ baselines may be less heterogenic in their values compared to the classic $\delta^{15}\text{N}$ tracer (see lines 184 to 190, 213 to 224 original manuscript)

In any case, we agree with the reviewer that our results will spur further research on $\delta^{66}\text{Zn}$ as a marine trophic level proxy, along with issues regarding $\delta^{66}\text{Zn}$ baselines. We also agree with the reviewer that our novel dataset has parallels with the early studies on $\delta^{15}\text{N}$, which despite their limitations (inherent in every proxy), turned out to be ground-breaking and changed the way we understand and study nutrient pathways and trophic interactions in past and present ecosystems.

During the rephrasing of our Results and Discussion section, we now address issues of baseline and other potential influences more clearly (lines 245-251, 331-342, 394-406) which we believe, combined with our state-of-the-art section in the Supplementary Information, addresses the issue raised by reviewer more clearly now. Please also see our response to comment 2.2c.

1.3

Based on the aforementioned comment, another main concern is about the goal of the paper and the “Implications for the megatooth shark extinction” (starting line 267). The interpretation is finally very speculative, and the strong statement “the $\delta^{66}\text{Zn}$ values in this study lend support for Carcharodon driving the extinction of *Otodus* through resource competition” (lines 275-275) is very hard to buy. I don’t see the evidence for direct competition between the *Otodus* and *Carcharias* lineages. I rather see niche partitioning.

We consider resource competition to still be a viable possibility because, while niche partitioning is generally the product of competition, the fact is that *Otodus megalodon* became extinct where our data show the decreased trophic position in *O. megalodon* relative to its predecessor *O. chubutensis* and the rise of *C. carcharias*. Nevertheless, we have significantly modified our discussion (as well as the title of our manuscript) on resource competition and have significantly softened our conclusions (demonstrated particularly in lines 394-406, 451-472).

1.4

I think the authors have made a focus on shark interactions, but it is overlooked that species interactions could have acted between unrelated clades, such as cetaceans. The first taxon that is ignored in this study is the killer whale (*Orcinus orca*). It could be also linked to the extinction of the megatooth shark. Given the fossil record of *Orcinus*, it appears that *Orcinus orca* is known from Japan (Hokkaido), in the Pliocene (Zanclan: 5.332-3.6 Ma). After that *O. orca* spread around the world since the late Pliocene. So, this taxon could have also played a role in the demise of the megalodon.

We focused on shark interactions in this manuscript for a good reason—i.e., shark teeth are said to be the most common vertebrate remains in the fossil record and the only fossil samples analysed herein. Nevertheless, the reviewer’s comment is a good one, and we have added *Orcinus* in the final paragraph as a potential competitor for food resources and thus a possible factor to *O. megalodon*’s extinction (lines 460-463). In addition, we have ensured that our manuscript would not imply that the emergence of *Carcharodon carcharias* is the sole reason for *O. megalodon*’s extinction. We believe these adjustments have enhanced the quality of our manuscript.

1.5

Another obvious unnoticed taxon could be the interaction with the extinct *Leviathan melvillei*. Indeed, the two giant marine predators, *O. megalodon* and *L. melvillei*, were found in the same mysticete-rich Serravallian locality (Lambert et al. 2010 – Nature) and studying their interaction might be relevant to understand the extinction of megalodon. Furthermore, large physeteroid teeth found in other Miocene localities worldwide indicate that giant raptorial sperm whales occupied a top predator position in various marine regions during the Miocene, a role now mostly taken by *Orcinus*. As such, the appearance of these sperm whales probably had a profound impact on the structuring of middle to late Miocene marine communities and food chains, like the impact of *Orcinus*. You would certainly argue that *L. melvillei* cannot be responsible for the extinction of the megatooth shark because of time difference between the two lineages (*L. melvillei* went extinct earlier), but you cannot reject the hypothesis that their possible interactions could have mediated the diet shift of the megalodon. Same rationale with *Orcinus*, which more actively predate mysticete while the great white shark is more a scavenger of dead whales.

This comment is also a good one, but we have ultimately decided not to add *Leviathan melvillei* to our discussion unlike *Orcinus* discussed above because *L. melvillei* is confined to the Southern Hemisphere, whereas all of our samples come from the Northern Hemisphere. Nevertheless, we anticipate that combining our shark-based data with $\delta^{66}\text{Zn}$ of predatory cetaceans, including *Orcinus* and *Leviathan melvillei*, would be very interesting for future studies, pending the availability of samples (e.g., besides permission for destructive sampling of fossils is often difficult to obtain because cetacean fossils are less common than shark teeth in the fossil record, securing modern cetacean samples have been difficult due to many strict marine mammal protection laws and regulations).

1.6

This is to say that the view presented here is very simplistic and incomplete. Therefore, the conclusions presented, including the title, are overstated. Their results rather point to niche partitioning rather than direct interactions that could have led to extinction of a lineage.

We are uncertain as to how niche partitioning could lead to extinction because niche partitioning is essentially a ‘coping mechanism’ for competition. However, we have significantly rewritten our conclusions by toning down our initial claims, including the title to better reflect the niche partitioning/overlap between sharks. In fact, our revised manuscript no longer focuses on *O. megalodon*’s extinction but rather on general *Otodus* and *Carcharodon* trophic ecology.

1.7

Some results are hard to believe, especially the fact that the mean $\delta^{66}\text{Zn}$ value for extant *C. carcharias* places the great white shark at a higher trophic level than the Atlantic *O. megalodon*. The authors acknowledge the limited sample size but then speculate on diet preferences of Pliocene great white and megatooth sharks.

We discuss in that paragraph our $\delta^{66}\text{Zn}$ data, which indeed indicate a higher trophic level of extant *C. carcharias* compared to Atlantic *O. megalodon* (Figure 3). Keeping with good scientific practice, we then mention limitations in samples size as a potential caveat to our interpretation. While we acknowledge the small sample size for modern *C. carcharias*, their results (which are from different locations) are very much in line with other high trophic level shark $\delta^{66}\text{Zn}$ values, both extant and fossil. We expand on this even more now (lines 394-406).

Furthermore, a higher trophic level of modern great white sharks compared to *O. megalodon* is very much in line with the current knowledge of their respective diets, which is a main discussion point of our manuscript. Modern great white sharks feed largely on piscivorous mammals, whereas most research into the diet of *O. megalodon* indicates that filter-feeding mysticetes were probably an important food item. Mysticetes are thus feeding on a much lower trophic level (mainly zooplankton and small fish) than piscivorous mammals such as seals and dolphins (and other prey items of the modern great white shark). The logical consequence is that a predator feeding more on mysticetes would have a lower trophic level (and higher $\delta^{66}\text{Zn}$ values) than a predator feeding more on toothed whales and pinnipeds like the modern great white (which would only scavenge mysticetes when available and otherwise feed on piscivorous smaller mammals and fish). We also see this when comparing the modern filter-feeding *Cetorhinus maximus* $\delta^{66}\text{Zn}$ values (+0.47 to +0.52 ‰) with those of various higher trophic level sharks feeding on a variety of higher trophic levels and preying on fish, cephalopods, and mammals (-0.71 to +0.28 ‰). While *O. megalodon* most likely also fed on piscivorous mammals and perhaps elasmobranchs and teleosts (otherwise, $\delta^{66}\text{Zn}$ values would likely be higher), our data implies that mysticetes were probably an important part of their diet (especially when compared to *O. chubutensis* and modern great white sharks).

In short, it is worth emphasising that just because *O. megalodon* was a much larger animal than the modern great white shark does not necessarily mean it fed on a higher trophic level. Such a view does not consider the trophic level of available and targeted prey species. We have significantly re-written the discussion to clarify this point (as also suggested by reviewer 3, comment 3.6) and we hope our adjustments are satisfactory (specifically in lines 427-450).

1.8

An important neglected aspect is that fossil occurrences inform on first and last appearance datum (FAD and LAD) of a taxon but given the various biases in the fossil record these FAD and LAD are not the 'true' ages that determine the species' lifespan. Based on an analysis of the lamniform fossil record (Condamine et al. 2019 - PNAS), the origin of *C. carcharias* is estimated at 6.76 Ma (95% CI: 5.45-8.59 Ma), while the origin of *O. megalodon* is estimated at 23.3 Ma (95% CI: 22.02-24.90 Ma). Importantly, the extinction of *O. megalodon* is estimated at 0.754 Ma (95% CI: 0.003-1.96 Ma), a bit more recent than the LAD in the

Zanclean. This suggests that the interactions between *C. carcharias* and *O. megalodon* could have lasted longer and that niche partitioning was an efficient way to avoid direct competition. This very same study showed diversity dependence within the large-sized lamniform species (including *C. carcharias* and *O. megalodon*) so that rates of species formation and extinction depend on the number of species of the same guild.

We thank the reviewer for pointing out Condamine et al.'s (2019) work. While the reviewer's point about the 'longer' interaction is possible, our study demonstrates a snapshot of the possible trophic relationship. We understand that 'species formation and extinction' are complex matters as the reviewer alludes, but the main goal of this paper was to examine the trophic ecology of extant and fossil sharks and whether the demise of *O. megalodon* could be explained by ecological factors (evidenced by this new proxy) such as the rise of *C. carcharias* through competition. Resource competition is now discussed much more conservatively as one of many possibilities in our revision, whereas the extinction of *O. megalodon* in general is not a main topic of the manuscript anymore. Nevertheless, we have added Condamine et al.'s (2019) paper to our discussion (line 461).

1.9

Why not using calcium (Ca) isotope values from tooth enamel (expressed as $\delta^{44/42}\text{Ca}$) to investigate resource partitioning? This has been done in theropod dinosaurs for instance (Hassler et al. 2018 – PRSB). Because Ca isotope ratios are widespread, it is possible to infer the ancestral diets (for instance, fishes for spinosaurs). This provides more clues than zinc isotopes, it seems.

We appreciate the comment, but in this study, we extensively explore for the first time the use of $\delta^{66}\text{Zn}$ values to investigate the trophic ecology of ancient elasmobranchs and to specifically investigate the feeding ecology of megatooth and great white sharks. We do, however, anticipate performing a combined analysis of $\delta^{44/42}\text{Ca}$ with $\delta^{66}\text{Zn}$ values on the same teeth trying to further refine the shark's diet and niche partitioning. However, Ca isotope analyses on marine reptiles and sharks so far demonstrate that the trophic level difference in $\delta^{44/42}\text{Ca}$ is less pronounced than in terrestrial food webs and high trophic level feeding aquatic vertebrate top predators such as marine reptiles have quite similar $\delta^{44/42}\text{Ca}$ values (Martin et al., 2017). Thus, the trophic level discrimination power is better with Zn isotopes.

Regarding the spacing of $\delta^{44/42}\text{Ca}$ isotopes and their use as ecological proxies in comparison to $\delta^{66}\text{Zn}$, we would like to indicate that the isotopic spacing for $\delta^{66}\text{Zn}$ in a single modern location (i.e., KwaZulu-Natal South Africa [KZN]) is 0.88 ‰ (only carnivorous elasmobranchs), whereas the total spacing among enameloid of all extant taxa in our study is 1.26 ‰. In comparison, the $\delta^{44/42}\text{Ca}$ spacing observed previously among extant elasmobranchs from various locations from zooplanktivores to apex predators lies at only 0.5 ‰ (Martin et al., 2015). Zinc isotopes, therefore, have a much larger trophic discrimination factor despite the lower isotopic variability of zinc in the earth's crust and its larger atomic mass compared to Ca (i.e., less fractionation potential). In addition, the Ca isotopic composition of tooth enameloid is controlled by both seawater $\delta^{44/42}\text{Ca}$ and trophic level (Akhtar et al., 2020), with estimates indicating that Ca transported across the gills in teleosts may make up to 70 % of the total Ca in their body (Sundell &

Bjornsson, 1988). As shown in our study, $\delta^{66}\text{Zn}$ values are likely exclusively controlled by diet, with Zn uptake by gills having no apparent effect on skeletal $\delta^{66}\text{Zn}$ values (along with Zn being highly depleted in marine water). Therefore, we argue that $\delta^{66}\text{Zn}$ is likely to provide more clues about diet in marine vertebrates than $\delta^{44/42}\text{Ca}$.

1.10

Finally, I would include a bivariate plot of $\delta^{18}\text{O}$ and $\delta^{66}\text{Zn}$ because the authors use results from both isotopes to infer potential preys (Zn for trophic position and oxygen to infer where they could have preyed within the water column)

We have only combined $\delta^{18}\text{O}_p$ and $\delta^{66}\text{Zn}$ values on the same tooth for 18 fossil teeth (restricted to Miocene Germany and Malta, see Supplementary Data 1). We decided against plotting both together as we felt this would have been criticised by reviewers (and rightly so). Oxygen isotopes were measured only on a fraction of the teeth for which we have $\delta^{66}\text{Zn}$ values and include two different sites. Such a plot would thus seem like cherry-picking data points (see Response Figure 1 below). Additionally, we use $\delta^{18}\text{O}_p$ of the few samples analysed mainly for diagenesis purposes (i.e., relative values among species) and in one instance to demonstrate the likely coastal habitat of *Carcharhinus* sp. from Miocene Malta, where we see a very clear difference in $\delta^{18}\text{O}_p$ between *Carcharhinus* sp. and sympatric species. We do not have enough data per species to infer where prey was hunted in the water column for all analysed species. As such, we believe our representation of the $\delta^{18}\text{O}_p$ values in Supplementary Figure 8 and Supplementary Data 1 is sufficient.

That being said, we do think that combining both proxies within a fossil assemblage with a larger sample set of both proxies analysed on the same teeth, has the potential to allow for a much more comprehensive picture of ancient ecosystems and feeding ecologies. We hope that our study will inspire such approaches in the future and we plan to perform similar studies ourselves (see also our response to 2.9).

Response Figure 1: Zinc versus oxygen isotopes measured on the same teeth from Miocene Germany and Malta. Note the very low number of teeth with both proxies measured per species, precluding a detailed isotopic niche interpretation with the current data set.

Minor comments

1.11

Lines 59-60: “likely provides better long-term preservation than organic collagen-bound nitrogen¹¹, the most commonly used isotopic trophic tracer to date”. What about calcium isotopes that have been used quite a lot, in dinosaurs for instance?

It is true, that Ca isotopes ($\delta^{44/42}\text{Ca}$) in enameloid are also likely to show a better long-term preservation than organic collagen-bound nitrogen. However, this study focusses on zinc isotopes, which we compare in this sentence to the most common (and reliable) trophic level indicator organic bound $\delta^{15}\text{N}$ in terms of preservation. Please also refer to our response to comment 1.9.

1.12

Lines 63-66: I found the presentation of the megalodon quite short. We know much more than what is said here. I don’t agree with the idea that its extinction remains an enigma.

We believe we have presented just enough background information about *Otodus megalodon* with appropriate references, where our revision now places more emphasis on introducing the general scientific problems related to palaeodietary reconstructions and introducing the novel zinc isotope proxy, rather than on *O. megalodon*’s extinction. We hope our revision satisfactorily addresses the reviewer’s concern.

1.13

- Figure 1: Why are the South African specimens separated from the rest? I guess what is really not clear to me is the grouping of the second box that includes species from the US, Island, Norway, etc.

The specimens from KwaZulu-Natal (South Africa, KZN) were grouped separately because it is these samples for which we have the highest number of extant elasmobranch species. Additionally, we discuss the KZN species separately in the results and discussion section to highlight the very high isotopic spacing (0.88 ‰) in the $\delta^{66}\text{Zn}$ values among the different sympatric elasmobranch species caught within a single geographic area. We agree with the reviewer though that it wasn't as clear as it could be. Thus, we have now changed the figure and inserted a dashed line between the KZN samples and the others.

As for the second box, we grouped the species according to increasing isotope values (but keeping specimens from the same locations together). In doing so, we have followed the suggestion made by this reviewer in the next comment 1.14. Dashed lines further separate different sampling locations.

1.14

- Figure 2: I would find the figure more visually informative if the species were arranged in decreasing or increasing order of their values.

We agree. This is why we did arrange the species by increasing values within each time period and location in our original manuscript and have not changed it since. A separation according to time period and locations is obviously important, as well as keeping species appear in the same order in each box were possible (see also our response to comment 1.13).

1.15

- Figure 3: What are the dark squares in *Otodus*? That's not explained in the legend

In the figure caption we explain that *Otodus* spp. represents a combination of both *Otodus* species, which in the figure have different colours (dark and light blue). We now also include the colour code in the caption to better clarify this.

Reviewer #2 (Remarks to the Author):

2.1a

This paper reports a study to examine trophic level distinctions amongst extinct shark taxa from the Miocene to the Pliocene based on the determination of Zinc isotopes ($^{66}\text{Zn}/^{64}\text{Zn}$) in shark tooth enameloid, having first examined these and other isotopic (carbon nitrogen, oxygen) compositions in extant shark's teeth. Zn isotopes are a rather new means for examining trophism, that to date have been applied in a very limited way in marine contexts. Shark teeth lend themselves to this application because their teeth are diagnostic to species, they are well-preserved over long periods of time, their structures

are well-understood, and their size reflects in some degree the size of the animal. The ultimate focus/hypothesis in this study is on distinctions in trophic level between the giant extinct megatooth shark (*Otodus megalodon*) and the (still extant) Great White shark (*Carcharodon carcharias*) that might have led to competition and eventual extinction of the former.

The authors first establish the 'credentials' of Zn isotopes by examining and comparing multiple isotopic and trace element systems of several extant shark taxa from different ocean regions. Zn isotopes are compared against carbon and nitrogen isotopes from associated organic components, showing that nitrogen isotopes, the "standard" trophic indicator in marine system, compare well, but it is more variable than Zn isotopes. From this it is deduced that Zn isotopes may provide a more globally standard indicator, although certainly this remains to be tested further. Tests of trace element distribution to assess diagenetic invasion of the fossil tooth enameloid are carried out, which identified areas to avoid. The stated similarity of phosphate oxygen isotopes in fossil versus extant teeth was not very convincing to this reviewer, from the little information we are provided about the palaeoceanographic contexts. It should be relatively simple to back up this similarity in present vs past ocean temperatures with hard data from the palaeoceanographic marine core literature. Still, I think that overall, the case is convincing. from phosphate any sort of influence on the results.

We thank the reviewer for the very thorough and insightful review and we appreciate that the reviewer comes to the same conclusion that our $\delta^{66}\text{Zn}$ data indicates a potential for a new global marine trophic indicator, and we agree (as also indicated in the manuscript) that this remains to be tested further. We believe our study will inspire additional research into $\delta^{66}\text{Zn}$ baseline variation in the marine realm, as the absence thereof could be of great significance for modern and past ecological studies.

Regarding our $\delta^{18}\text{O}_p$ results, we see now that we could have phrased them differently and more clearly to explain our approach. We only use the $\delta^{18}\text{O}_p$ values in the manuscript in a relative context, i.e., we compare the $\delta^{18}\text{O}_p$ of different Miocene species to each other to infer their likely habitat. Specifically, we do this to highlight that $\delta^{18}\text{O}_p$ indicate similar relative $\delta^{18}\text{O}_p$ values to each other and, therefore, relative temperature ranges as observed in modern equivalents from KZN (Vennemann et al., 2001). This is used as an indication of generally well-preserved enameloid in a diagenetic context and to discuss the ecology of *Carcharhinus* from the Miocene of Malta.

We have rephrased the appropriate text to mention $\delta^{18}\text{O}_p$ values reflect "species-specific relative in-vivo temperature ranges as expected compared to the habitat use of equivalent modern species" (lines 318-319).

2.1b

Zn and other isotope analyses are then carried out on a range of extant taxa of various sizes, typical prey, and from different oceanic regions, and statistically compared. Given that this is a very complex dataset the rationale for using largely pairwise statistical comparisons for each isotope and species was not clear (nor were the results clear). It would have been preferable to present the data using the tools commonly used for isotopic niche distinctions amongst ecologists, for instance Standard ellipse areas (SEAs) for

binary isotope systems, or even NicheROVER (Swanson et al. 2015) for multi-dimensional systems. These allow both statistical comparisons as well as far better visualisation of the data.

The current study used Tukey and Games-Howell's post-hoc pairwise comparison test to examine the means of more than two groups (i.e., a categorical variable presenting more than two levels). Following the one-way ANOVA omnibus analysis to test the null hypothesis (whereby samples in all groups are drawn from populations with the same mean values), follow-up comparisons are required to identify and specify which groups are different from one to another.

Specifically, the omnibus one-way ANOVA test (and consequently post-hoc pairwise comparison test) was chosen and conducted since the current study primarily focuses on exploring $\delta^{66}\text{Zn}$ variability in fossil shark species but also because of methodological reasons. Other numerical response variables are presented ($\delta^{18}\text{O}_p$, $\delta^{13}\text{C}$, $\delta^{15}\text{N}$, [Zn], etc.), but using them in conjunction with $\delta^{66}\text{Zn}$ values in statistical analyses is ill-advised. Indeed, some of them (i.e., $\delta^{18}\text{O}_p$, $\delta^{13}\text{C}$, and $\delta^{15}\text{N}$) are only available for small sample subsets as they were either used to explore additional questions, such as diagenesis investigation (i.e., $\delta^{18}\text{O}_p$), or simply not readily available throughout the full dataset. Specimens presenting values for such numerical response variables are thus inadequate for multi-proxy statistical analyses as they carry inherent biases (e.g., only available for specific species, localities or feeding habitat, insufficient sample size, etc.). For all intents and purposes, only a single numerical response variable, $\delta^{66}\text{Zn}$ values, was thus suitable for statistical analyses without incurring unrepresentative results.

For similar reasons, multiples omnibus ANOVA tests were performed (i.e., between targeted species, periods, or localities), rather than two-way ANOVA or similar tests, as a means to better tailor the scientific questions (and comparisons) under investigation. For example, to investigate the homogeneity/heterogeneity of $\delta^{66}\text{Zn}$ values across the globe, one should at least control for: species (as different diets could otherwise influence the results), period (as the results could otherwise be influenced by different baseline, if applicable), and sample size (as unequal sample size generally hinders robustness and even feasibility of many statistical analyses).

In contrast, to what was stated by the reviewer, pairwise comparison tests were performed solely on the Zn dataset. As mentioned above, the other isotopes used in this study were used to test specific questions and underwent appropriate tests specifically. As for nitrogen, correlation tests were conducted to ascertain the trophic position obtained with Zn stable isotopes against a well-established trophic level tracer. In both cases, many specimens come from different localities and habitats or exhibit different feeding habits, making direct comparisons between specimens ill-advised. Therefore, the use of Standard ellipse areas (SEAs) for binary isotope systems and NicheROVER (Swanson et al. 2015) for multi-dimensional systems is problematic in our case.

For fossil samples, we simply lack proxies to compare with, as we only have $\delta^{66}\text{Zn}$ data for the vast majority of samples. The only 18 samples with both $\delta^{66}\text{Zn}$ and $\delta^{18}\text{O}_p$ data are also distributed among two fossil sites, further complicating a direct comparison (see also our response to 1.10).

For extant samples, we have $\delta^{15}\text{N}$, $\delta^{13}\text{C}$, and $\delta^{66}\text{Zn}$ data for most of our samples. However, the problem here is that we investigate teeth from a variety of globally distributed oceanographic areas. Therefore, comparing niche size and habitat will be problematic and prone to error using NicheROVER (see also Swanson et al., 2015, chapter "*Cautions and future directions*"). We only have a handful of samples from

the same sites and most extant teeth come from a variety of geographic locations (11 locations) distributed around the globe. In these different oceanographic regions, isotopic baselines will likely vary (perhaps more for $\delta^{15}\text{N}$ and $\delta^{13}\text{C}$ than $\delta^{66}\text{Zn}$). Further complications can include differences in resource availability, community composition, among others. Either performing Standard ellipse areas (SEAs) or NicheROVER will thus **not** yield any meaningful results related to isotopic niche distinction in our case.

We combined $\delta^{15}\text{N}$ and $\delta^{13}\text{C}$ with $\delta^{66}\text{Zn}$ on the modern samples for multiple reasons. These include:

- To consider factors other than diet on enameloid $\delta^{66}\text{Zn}$ values (e.g., original manuscript lines 180 to 183)
- To test the response of all three dietary proxies on controlled fed versus wild fed fish (e.g., original manuscript lines 146 to 151)
- To compare $\delta^{66}\text{Zn}$ values with $\delta^{15}\text{N}$ and FishBase trophic levels to document trophic fractionation: We acknowledge the impact of baseline on $\delta^{15}\text{N}$ and perhaps $\delta^{66}\text{Zn}$ among locations and the uncertainty in FishBase trophic levels, which are likely underestimated and may vary among locations (lines 161-163 and caption of Supplementary Figure 3). The fact that we still get a meaningful correlation between $\delta^{66}\text{Zn}$ of FishBase trophic levels and, though weaker, $\delta^{15}\text{N}$ and FishBase trophic levels demonstrate a general decrease in $\delta^{66}\text{Zn}$ values with trophic level. A comparison with more precise trophic levels (i.e., not limited to maximum trophic level 4.5) is likely to demonstrate an even higher correlation for $\delta^{66}\text{Zn}$ values. Likewise, a larger number of species tested from the same location (and corrected for $\delta^{15}\text{N}$ baseline between coastal and open marine) is likely to produce a higher correlation between $\delta^{66}\text{Zn}$ and $\delta^{15}\text{N}$, although this remains to be tested and is obviously, beyond what is possible for this data set.

Nevertheless, we agree with the reviewer that performing these analyses on multiple marine species from the same location would be very useful to the ecological and palaeontological communities. Following our pioneering study, we are currently developing studies that will include multiple isotope proxies with the use of analyses such as SEAs and NicheROVER in mind. These will be done on multi-proxy isotope data from the same region for both present and past assemblages. Here we also like to stress that, for this study, we were not able to combine a large variability of extant shark teeth from a single location due to the many strict protection laws and regulations and difficulties of gaining access to modern material. However, our samples were sufficient for our goal to demonstrate the reliability of $\delta^{66}\text{Zn}$ as dietary/trophic proxy in fish and to document the variability in $\delta^{66}\text{Zn}$ values.

To clarify, as to why including SEAs or NicheROVER on either the extant or the fossil teeth will not reveal any ecologically relevant information, we have made the following adjustments to the manuscript:

- We highlight the information that extant teeth were selected from a variety of different oceanographic locations even more now in lines 85-86 in the introduction and in lines 183-185, 193-195, 207-208, 211-213, and 246-251 when discussing the results of extant $\delta^{66}\text{Zn}$ values.
- To further clarify that $\delta^{18}\text{O}_p$ was only measured on a subset of teeth from Miocene Germany and Malta, we now include a statement in the Methods section (lines 578-579) and highlight the information in the discussion (line 316).

2.2a

There are two other areas of concern. First, it is not at all clear that the main hypothesis, that the giant extinct megatooth shark (*Otodus megalodon*) was outcompeted by the Great White shark (*Carcharodon carcharias*) can be tested using this data, and given gaps in the information provided. The Zn isotope data do show some differences between the two taxa but these differences vary according to both time and oceanic region. The hypothesis is too broad, and other possibilities are not entertained. Fig 2 for instance shows significant differences in Zn isotope composition for *O. megalodon* in the early Pliocene of Japan (lower values = higher trophic level) vs same period for North Carolina. It's not clear how the conclusion is reached that *Carcharodon carcharias* outcompeted *Otodus megalodon* based on this data. There might be, for instance, regional oceanic conditions (currents?) and regional prey species that differ.

We now discuss the aspect of competition between *O. megalodon* and *C. carcharias* in a much more conservative manner, hinting at the possibility of competition while also mentioning alternatives to interpret our data such as niche partitioning (specialisation on different prey with a similar trophic level, lines 452-454). Our interpretation comparing *O. megalodon* and *C. carcharias* now also focusses on the North Carolina site, where both species occur sympatric with a similar range of $\delta^{66}\text{Zn}$ values (lines 451-459). Our direct comparison of these species in the Pliocene is thus independent of baseline (regional environmental conditions) and regional prey species differences. We believe our adjustments have adequately alleviated the reviewer's concern.

2.2b

Overall, oceanographic contextualisation is very sparse in this study. This is a problem for several reasons, including some recognition of ocean currents and temperatures, in relation to productivity and the potential array of prey species. Because the oceanography is not clear and not referenced or explained, the conclusion about similar temperatures derived from PO_4 oxygen isotopes in past and present could be challenged.

We do not directly compare $\delta^{18}\text{O}_p$ values from the past to those of the present at any point in the manuscript. We see now that this misunderstanding comes from the fact that we stated that $\delta^{18}\text{O}_p$ values are in the same range in the Miocene as today for the KZN sharks from Vennemann et al. (2001). While this is a correct statement and only used in the results, it gave the wrong impression that we directly compare modern to fossil $\delta^{18}\text{O}_p$ values to quantify palaeotemperatures. In our revised manuscript, we now specify more clearly that we only compare the species-specific relative $\delta^{18}\text{O}_p$ values from the European Miocene sites; i.e., higher temperature for coastal *Carcharhinus* compared to more deep-water fauna like *Pseudocarcharias* and *Mitsukurina* (see also Supplementary Figure 11) with no difference between both European Miocene sites when analysing the same taxa (see also our response to comment 2.1). This is in line with relative species-specific temperature ranges observed in modern equivalent species for the KZN by Vennemann et al. (2001). Absolute temperature comparisons between the European Miocene and recent KZN are irrelevant for our study. Please also refer to our response to 2.2c below for more on the oceanographic contextualisation.

2.2c

There is also the question of the influences of continental dust on ocean systems, depending on where in the ocean. We know that continental dust is far-reaching and influences ocean productivity (for instance Saharan dust across the Atlantic), but further, several studies have shown that Zn isotopes can be used as tracers for the dust clouds since Zn varies isotopically by source. In other words, there are other possible ways in which Zn isotopes might vary by region. Such issues need addressing if the conclusions drawn on the basis of comparisons from different periods and places can be accepted with confidence.

While we agree that multiple factors could explain baseline $\delta^{66}\text{Zn}$ variability for a given site, our data shows strong agreement in mean values across geographic areas, in line with the only previous Zn isotope publication that can be used for comparison (McCormack et al., 2021). We acknowledge that baselines for Zn are largely unknown, mainly because of the novelty of this proxy particularly in the marine realm. Nonetheless, we provide a summary of dissolved marine $\delta^{66}\text{Zn}$ and biological fractionation in the Supplementary Information. It should be stressed, however, that baseline variations seem less important for $\delta^{66}\text{Zn}$ than for example for $\delta^{15}\text{N}$ (as also acknowledged by this reviewer in a previous comment 2.1a).

The problem of baseline variation is also the reason as to why we focus not solely on *Otodus* and *Carcharodon* teeth, but instead present multiple species from each fossil location and period where possible. This allows us to draw baseline independent conclusions by comparing the relative spacing of $\delta^{66}\text{Zn}$ values within assemblages (e.g., the high trophic level of *Otodus chubutensis*, the low and similar trophic level of *Carcharhinus*). We can then further compare, where possible, taxonomic groups with each other across assemblages and periods (which we do). These comparisons of taxa like *Galeocerdo*, *Carcharhinus*, *Otodus chubutensis*, and *Carcharias* indicate that baseline variations are most likely negligible relative to the trophic level fractionation effects.

We therefore contend that strong baseline variations seem unlikely to explain the observed variability in our samples; though we do acknowledge this potential factor limiting our interpretations repeatedly (e.g., lines 216 to 218, 259 to 266, original manuscript and lines 331-342, 394-406 tracked changes manuscript). We cannot fully exclude baseline variations for the Japanese *O. megalodon* samples as we have no other taxa for that site, but we again acknowledge this fact (lines 259 to 266, original manuscript, see also response to comment 2.2a) and focus our discussion on trophic levels of the better constrained Atlantic populations.

Our results therefore imply that discussions on possible differences in baseline due to external factors and a lengthy introduction on different oceanographic contexts are redundant. Still, in the following we briefly entertain this line of discussion, specifically in relation to dust input to the ocean: Most terrestrial geological materials have $\delta^{66}\text{Zn}$ values close to +0.3 ‰ typically in ranges of +0.2 to +0.6 ‰ for most igneous rocks, clay minerals, loess and general eolian dust, shales and most carbonates (Cloquet et al., 2008). Although some ore deposits and marine carbonates may show a wider range of values. The reasons for this low isotopic variability is the single oxidation state of Zn (+2) and the low capacity of igneous processes to fractionate Zn isotopes (Cloquet et al., 2008). Therefore, worldwide general $\delta^{66}\text{Zn}$ input into the ocean should be relatively homogenous in value, though there are few studies to draw from. We are aware of recent publications discussing the impact of anthropogenic aerosols as a source of dissolved Zn

in marine surface waters (e.g., Liao et al., 2020). These studies, however, have no meaningful bearing on our fossil results (which mimic in absolute value those of modern taxa) that were potentially only influenced by geogenic mineral dust input to the oceans.

The systematics of Zn isotope fractionation in marine surface water, the reasons for the globally homogenous $\delta^{66}\text{Zn}$ deep water and bioavailability of dissolved Zn are all subject of intense debate. Very few studies provide information about $\delta^{66}\text{Zn}$ values of dissolved Zn in marine surface water, which may also vary seasonally, with dissolved Zn not necessarily being bioavailable and uncertainties regarding fractionation of Zn by primary producers. Discussing reasons for baseline variations (for which we have no data) and potential factors controlling them on highly mobile species that may integrate multiple food chains preying upon potentially highly mobile prey and then applying that to the fossil record is highly speculative at best. Not to mention, that again we clearly indicate baseline variation likely has little effect on our $\delta^{66}\text{Zn}$ data (and thus effects of mineral dust input, which is likely isotopically relatively homogenous compared to biological material), evidenced by the high spatial and temporal coherence within a given taxon/diet (e.g., *Galeocerdo*). We acknowledge that further studies into baseline variation or lack thereof will be necessary moving forward, but these are way beyond the scope of this first study of $\delta^{66}\text{Zn}$ in modern and fossil non-mammal marine vertebrates.

2.3

The next issue is of a different sort. This paper seems (to this reviewer) to consist of two natural sections, each of which has a large amount of complexity and test, but putting them together makes it rather difficult to see the wood for the trees, so to speak. The section on extant species and how Zn isotopes behave for different taxa, conditions and prey, and in relation to the other main trophic indicator, nitrogen isotopes, is of course essential fundamental information to tackle the fossil issues, but on its own it still has some important contributions for modern shark ecology or more broadly, marine trophic systems (the nitrogen vs Zn isotope differences for instance). But the complexity of the many comparisons amongst numerous taxa in different places and conditions, and their tests for significance, make it very hard for a reader to distinguish the important features. The authors might consider reporting first their modern data? In doing so they also use some of the tools developed by ecologists for niche breadth comparisons within ecosystems such as Standard Ellipse Areas (SEAs) for binary isotope systems or better still the new NicheROVER (Swanson et al. 2015) for multi-dimensional systems. That would be really useful to both the ecological and palaeontological communities. A stronger palaeoceanographic context would provide a more solid basis for interpretation of the fossil data .

We agree with the reviewer that our $\delta^{66}\text{Zn}$ data on the extant fauna in itself already offers an important contribution to modern ecology with important implications on the future of trophic ecological studies and beyond. While we certainly could have made two or more manuscripts out of our data set, we ultimately decided to submit it as a more complete and comprehensive story. Naturally, such a decision is a matter of taste. Still, we have added a short introduction to the new results and discussion section to highlight the important results and aid the reader in following the subsequent discussion (lines 182-190).

We fully agree with the reviewer, that a study combining the traditional $\delta^{15}\text{N}$ and $\delta^{13}\text{C}$ with $\delta^{66}\text{Zn}$ values on multiple marine species from the same location would be very useful to the ecological and

palaeontological communities. For such a study, applying Standard Ellipse Areas (SEAs) for binary isotope systems or NicheROVER (Swanson et al. 2015) for multi-dimensional systems will be incredibly useful. We plan to do such a study ourselves in the future based on the encouraging results of this pioneering study. We also anticipate our study to inspire other research groups to begin similar studies. However, we feel that with the available data from this first study such statistical analyses are unpractical for now (see our response to 2.1b for a detailed explanation).

Minor comments.

2.4

Repetition of the word "iconic" was somewhat jarring.

We have deleted the second iconic and with the restructuring of the abstract we now do not use the word at all.

2.5

Statistical analysis (Line 400 onwards) Analysis of variance (ANOVA) was performed to determine statistical differences in $\delta^{66}\text{Zn}$ values across fossil assemblages or among fossil and extant species but the text only mentions the fossils? Is this just a sentence error?

We understand that this sentence can be misread and have changed it accordingly. It now states: "Analysis of variance (ANOVA) was performed to determine statistical differences in $\delta^{66}\text{Zn}$ values across fossil assemblages, among fossil species and to compare fossil to extant species".

2.6

the first sentence of the abstract "The demise of the iconic gigantic Neogene megatooth shark, *Otodus megalodon*, remains enigmatic, ..." does to make complete sense. You mean the "reasons for the demise"?

With the restructuring of the abstract this sentence was omitted.

2.7

also in the abstract, the statement "Our $\delta^{66}\text{Zn}$ values are consistent with the hypothesis that *C. carcharias* outcompeted *O. megalodon*." is not supported by the data provided, would need more careful wording.

Done. We have changed the abstract substantially and now word our results in regards to *O. megalodon's* extinction much more carefully.

2.8

Line 56: "Studies to date indicate that $\delta^{66}\text{Zn}$ values successively decrease with increasing trophic level 9-13". Its worth pointing out that v few of them have been applied in marine, and then mostly in quite circumscribed archaeological contexts.

Done. We included a sentence highlighting that previous $\delta^{66}\text{Zn}$ studies are very few in the marine realm and limited to archaeological material. We also provide references to the only two previous publications on marine $\delta^{66}\text{Zn}$ data in an ecological context (lines 76-77).

2.9

Line 77. "by measuring tooth phosphate oxygen isotopes ($\delta^{18}\text{O}_p$)". Was this done on all the extant and fossil teeth, not clear though I may have missed?

Thank you for pointing out that this information was not clear. Although we do mention that $\delta^{18}\text{O}_p$ was measured for the European sites only (lines 201-204 original manuscript and Supplementary Figure 11), we have now also included this information in the methods section (lines 578-579) and explain in the discussion that it was only measured on a subset of teeth from the European Miocene sites (line 316), to make the information more accessible.

2.10

Line 97. "We also observe a correlation between trophic level and $\delta^{15}\text{N}_{\text{coll}}$ values ($R^2 = 0.42$, $p = 6.97\text{E}-6$, $n = 40$)". Bit surprised that its quite a weak correlation, is it because of the odd pesky outlier? Quite useful to id the outliers esp. since you conclude (rightly) that nitrogen isotopes are more dependent on local conditions.

This comment highlights that we could have been clearer in how we stated that the modern samples of this study were taken from various locations worldwide. We have now changed that by including this information to the introduction (lines 85-86) and results and discussion (lines 183-185, 193-195, 207-208, 211-213, and 246-251). As such, we believe the reviewer will now not be so surprised about the low correlation between $\delta^{15}\text{N}_{\text{coll}}$ and FishBase trophic levels; the reviewer indicates in this comment that $\delta^{15}\text{N}$ will depend on local conditions. In fact, we were quite surprised that the correlation was as good as it is given that $\delta^{15}\text{N}$ baseline is likely to vary substantially between the different marine settings.

2.11

Line 138/139. This statement "We observe no correlation between the total body length of *Otodus* spp., *Carcharodon hastalis* and *C. carcharias* and their respective $\delta^{66}\text{Zn}$ values.." is also surprising especially since multiple marine studies show that marine animals tend to consume larger (and hence higher trophic level) prey as body size increases. Do the uncertainties in estimating body length interference with this analysis?

This comment made us realise that we have provided too little information in that particular sentence. We thank the reviewer for pointing this out. The problem here is not uncertainties in body length. All teeth, for which total length estimations were possible, had already surpassed the body size at which dietary shifts likely occur. This means we simply lack teeth from juveniles that would likely indicate a different diet. We have added a sentence to the manuscript to clarify the lack of juveniles (lines 414-416).

2.12

Line 175. "Zinc isotope variability among marine organisms and their tissues is largely unknown, currently limiting our ability to identify specific food items based on shark enameloid $\delta^{66}\text{Zn}$ values beyond generally observed trophic level effects" . Might this be another basis for being more cautious in the conclusion that *O. megalodon* was outcompeted by *C. carcharias*?

We thank for the reviewer's thoughtfulness but disagree on this occasion. In this context, we specifically refer to *Carcharhinus* $\delta^{66}\text{Zn}$ values, which appear to indicate a much larger trophic level difference to sympatric species as when compared with $\delta^{15}\text{N}$ values. Indeed, *Carcharhinus* $\delta^{66}\text{Zn}$ values are often positive which is rarely found in other sharks indicating lower trophic levels. Noteworthy, previous studies have placed *Carcharhinus* on a similar trophic level as, for example, *G. aduncus*. We thus entertain here the possibility that unknown food sources, notably of a coastal, demersal or benthic or terrestrial baseline influenced, could cause the higher *Carcharhinus* $\delta^{66}\text{Zn}$ values. This mechanism is however not applicable to *O. megalodon* and *C. carcharias* (which have a typical range of pelagic shark $\delta^{66}\text{Zn}$ values).

Notably, we come to the conclusion that a specific unknown food item is likely **not** the cause of the higher $\delta^{66}\text{Zn}$ values. Neither is a terrestrial influenced baseline zinc source, when comparing $\delta^{13}\text{C}$ values among the KZN sharks (see lines 180-183 original manuscript and Supplementary Discussion lines 268-283). It is worth noting that while we cannot completely rule out the possibility that differences between a neritic- and oceanic-based diet might skew $\delta^{66}\text{Zn}$ -based trophic level determinations, differences between neritic and oceanic $\delta^{15}\text{N}$ baselines are well documented, e.g., with higher $\delta^{15}\text{N}$ values for neritic compared to oceanic zooplankton in the Gulf of Mexico (e.g., Dorado et al., 2012).

Still, in accordance with the reviewers suggestion, we discuss the potential of resource competition between *O. megalodon* and *C. carcharias* more carefully now.

2.13

Line 243. "For the *Carcharhinus* teeth from Malta, this interpretation is supported by lower $\delta^{18}\text{O}_\text{P}$ values than sympatric species, indicating a higher water temperature or lower salinity: i.e., a shallow and/or brackish water habitat (Supplementary Figure 11)." Where in time in relation to the Messinian crisis?

The lower $\delta^{18}\text{O}_\text{P}$ values in *Carcharhinus* relative to sympatric species from Malta indicate a more coastal habitat usage, similar to the here-studied extant *Carcharhinus* species which also have distinct $\delta^{66}\text{Zn}$ values. The teeth analysed herein are from the Burdigalian, which predates the Messinian salinity crisis by several million years. The Messinian salinity crisis is thus irrelevant for our data.

2.14

Line 283. "However, during the early Pliocene, the *Otodus* lineage represented by *O. megalodon* shows a considerable increase in the mean $\delta^{66}\text{Zn}$ value for the Atlantic populations, hinting at a reduced trophic position for the megatooth shark lineage in the Atlantic." Where about in the Atlantic? See comments on Saharan dust, know to be active from Miocene onwards.

We indicated multiple times in the original manuscript [(e.g., lines 120, 258, 312); Figure 2 and Supplementary Information and Supplementary Tables 3, 7 and Supplementary Data] that Atlantic *O. megalodon* teeth were recovered from Pliocene fossil sites in Florida and North Carolina. However, adult *C. carcharias* is typically migratory (Compagno, 2001) and *O. megalodon* may have been too. As our data indicates that $\delta^{66}\text{Zn}$ values may be less dependent on baseline variation, we see no reason to discuss a potential highly speculative effect of Saharan dust on the *O. megalodon* $\delta^{66}\text{Zn}$ values. With our current data, such a discussion would be counterproductive given that our results point to an absence of significant baseline variation for $\delta^{66}\text{Zn}$. Adding to that the fact that *O. megalodon* may have been migratory makes such a discussion even more speculative (see also our comment to 2.2c).

2.15

Line 293. "The extinction of *Otodus megalodon* could still be caused by multiple factors, including climate change³⁴. However, the apparent synchronous global occurrence of *Carcharodon carcharias* with the extinction of *O. megalodon*¹⁵". This is rather posthoc and not convincing.

As we have reorganised this part of the discussion, this sentence no longer exists in this form.

2.16

Line 351. "bone"? first mention of bone, this is a very different element and in sharks?

Bone collagen was taken for organic carbon and nitrogen isotope analysis on teleosts not sharks. We added "teleost" before bone to avoid further misunderstandings.

Supplementary I.

2.17

Line 45. "Heavy Zn isotopes tend to be partitioned into stiffer bonds ". Rather inelegant, can be improved Done. We changed it to "Heavy Zn isotopes are concentrated in stiffer bonds" (line 46 in the Supplementary Information tracked changes document).

2.18

Line 65 onwards. "The isotopic composition of marine dissolved Zn below 500 m seems to be globally homogenous with values close to +0.5 ‰, despite variable Zn concentrations^{14,15}. Consequently, the bulk isotopic composition of dissolved marine Zn is enriched in ⁶⁶Zn relative to its major inputs from rivers and aeolian dust, which centre on the global crustal average of +0.3 ‰^{16,17}." This is rather ignored in the main paper where very little information about where samples found and the surrounding ocean conditions.

In contrast to the reviewer's assessment, we believe we have given sufficient information where samples were found, as this is written in the text whenever we refer to specific samples. Additionally, Figures 1

and 2 indicate the origin of each extant and fossil sample, which is also given in more detail in the Supplementary Information and Supplementary Data 1.

Regarding the surrounding ocean conditions, please refer to our response to comment 2.2c.

2.19

Line 284. "Another possible explanation for the $\delta^{66}\text{Zn}$ differences observed between the KZN shark species could be taxon-dependent diet-bioapatite Zn discrimination factors." Far-fetched possibility! rather consider the conditions in that area and prey species.

We agree. The rest of that paragraph and the following paragraph discuss reasons as to why we argue that taxon-dependent diet-bioapatite Zn discrimination factors are an unlikely explanation for differences observed between *Carcharhinus* and other KZN shark species. Nevertheless, we feel that, since this is the first study of its kind, it is important to entertain different possibilities for the observed $\delta^{66}\text{Zn}$ variability. Ultimately, we agree with the reviewer (as also clearly stated in the main text) that dietary/trophic level differences are the most likely explanation. We discuss the possibility that a preference of demersal/benthic coastal prey for *Carcharhinus* species compared to more pelagic resources in the other KZN sharks may contribute to the observed differences. In any case, our $\delta^{66}\text{Zn}$ data clearly demonstrates different dietary resources for both present and past *Carcharhinus* species compared to most sympatric species.

2.20

SI Fig 2. What does "bulk teeth" mean? Did I miss an explanation somewhere?

Thank you for pointing this out. We have added the following statements to the methods section (lines 528-531 and 564-566): "As teeth of extant *Raja clavata*, *Scyliorhinus canicula* and *Etmopterus spinax* were too small to separate enameloid from dentine, we measured 20 to 30 complete teeth of a single individual together and report the results here as bulk teeth." and "Due to the small tooth size of *Etmopterus spinax*, we measured jaw cartilage and complete bulk teeth from this individual (Supplementary Data 1)."

Reviewer #3 (Remarks to the Author):

3.1

This new study by McCormack et al. tests the hypothesis that competition between the extinct megatoothed shark *Otodus megalodon* was outcompeted by the modern great white shark *Carcharodon carcharias* during the Pliocene through the analysis of Zinc isotope ratios from a wide variety of modern and fossil shark and fish teeth. Zinc isotopes have not been previously studied in nonmammals, and this method is a creative and novel approach to reconstructing ancient food webs. Critically, this study has produced a surprisingly large volume of zinc isotope data to establish a baseline for modern species (captive and wild), and contemporary as well as stratigraphically separated shark specimens. I am not a

geochemist, but nevertheless find the statistical results convincing. I am quite impressed with this study, and only have minor comments. Sincerely, Robert W. Boessenecker, Ph.D., College of Charleston, South Carolina, USA

We thank the reviewer for his appreciation of our work and his fair and constructive comments.

3.2

1) Are any of the fossil specimens phosphatized (e.g. blackened/mineralized and/or dense)? Shark teeth from the Lee Creek Mine and Bone Valley Formation (sources of some of the analyzed specimens) are often phosphatized. Based on discussions about uptake in heavy metals, the early diagenetic mineralization of shark teeth might similarly affect zinc isotope ratios.

We agree with the reviewer that such mechanisms and the possibility of Zn uptake must be explored, which is exactly why we discuss them in such detail in both the main manuscript and Supplementary Information.

In all locations analysed, the degree of phosphatisation and taphonomic wear on these specimens was variable. However, as clearly seen in our $\delta^{66}\text{Zn}$ values we observe no diagenetic impact on the Zn isotope signal, including no variation in $\delta^{66}\text{Zn}$ values with colouration, degree of phosphatisation or taphonomic wear. This is also clearly shown in Supplementary Figures 6, 12 and 13. Supplementary Figures 12 and 13 highlight how diagenesis affects zinc concentrations and $\delta^{66}\text{Zn}$ values (discussed in the main text and in Supplementary Information 3.2 in detail). Dentine and enameloid sampled along larger fractures demonstrates a reduction of Zn concentrations and an increase in $\delta^{66}\text{Zn}$ values. Diagenetic alteration of bioapatite $\delta^{66}\text{Zn}$ values in these sites is likely due to Zn^{2+} replacement by Ca^{2+} (and/or other cations) with a preferential replacement of the lighter Zn isotopes. This is in line with general equilibrium isotope fractionation considerations, our observations of fossil dentine (with sometimes even higher $\delta^{66}\text{Zn}$ values than the sediment), and the very low Zn concentrations in the embedding sediment of all sites compared to the very high Zn concentrations (and isotopically distinct Zn) in extant and fossil enameloid. As clearly observed in Supplementary Figure 6 as well as Figure 2, both Zn concentrations and $\delta^{66}\text{Zn}$ values in all fossil teeth analysed fall within the same range as observed for modern teeth of the same taxon. Our data thus demonstrates that likely all samples show no signs of diagenetic alteration of their pristine $\delta^{66}\text{Zn}$ values. Therefore, Zn uptake during diagenesis can be clearly excluded for the fossil sites we investigated herein.

To clarify that Zn uptake is unlikely; we now more clearly indicate that rather than Zn uptake any diagenetic alteration (mainly affecting dentine) is likely to be a Zn replacement in the Supplementary Information (lines 351-353).

3.3

2) The Pliocene-Pleistocene marine megafaunal extinction has not been dated, and post-dates the extinction of *O. megalodon*. The study by Pimiento et al. (2017) is a great one, but their faunal analysis used Pliocene and Pleistocene time bins – necessary given the coarseness of the available data – and the contrast in taxa between these two time bins artificially assigns the Pliocene-Pleistocene boundary as the

time of the extinction. However, it could be late Pliocene or early Pleistocene (~800 ka to 2.5 Ma), and it is possible that the extinction is asynchronous in different ocean basins. It's also possible that it was a period of more rapid faunal turnover rather than a classic example of an extinction event. See Boessenecker (2013: *Geodiversitas*: 35:4:815-940) for more on marine mammal extinctions/faunal turnover. Regardless, thanks to late Pliocene marine vertebrate assemblages, it's clear that the extinction of *O. megalodon* pre-dates this event by at least a million years or so (Boessenecker et al., 2019).

The reviewer has brought up many interesting plausible ideas concerning the possible patterns and timing of marine megafaunal extinction. However, we believe the examination and discussion about the marine megafaunal extinction to be beyond the scope of our paper, besides the “the coarseness of the available data” as the reviewer noted. In any case, restructuring of our discussion has led to the elimination of the sentence referring to the megafaunal extinction from the manuscript.

3.4

3) It is critical to note that while interesting, the studies on bite marks are qualitative (e.g. “we found some bite marks on this whale bone, how neat”) and a quantitative study of bite marks currently does not exist. Ideally, some study out there would compare presence/absence of bite marks for hundreds of specimens from stratigraphically separated assemblages and compare changes in trace sizes and prey taxa composition through time.

We fully agree. This is exactly the reason we believe our study to be so important for future palaeontological studies, as isotopic studies will more reliably reflect the overall diet than occasional fossil evidence, such as bite marks, which are not more than qualitative snapshots of faunal interactions. We discuss this also in the first paragraph of our Introduction. Nevertheless, bite marks can be quite useful in our context, as they substantiate that these faunal interactions, in this case between Neogene *Carcharodon* and *Otodus* with mysticetes as prey/food items, indeed occurred (though bite marks will not tell us if these happened regularly and rarely reliably indicate scavenging versus predation). Still, such bite marks corroborate our $\delta^{66}\text{Zn}$ data, indicating that Neogene *Carcharodon* and *Otodus* likely, though not exclusively, preyed upon mysticetes (i.e., lower trophic level).

We also think a more quantitative bite mark study as suggested by the reviewer would be amazing, especially if coupled to isotopic indicators such as zinc in shark teeth of the same assemblages. Unfortunately, this is way beyond our capabilities for this study.

3.5

241-3: this suggests that *Galeocerdo* spp. trophic level did not change despite an increase in body size – perhaps this is worth an extra comment.

Thank you for pointing this out. This is indeed worth mentioning, especially as tooth morphologies are so similar between *G. aduncus* and *G. cuvier* in line with our $\delta^{66}\text{Zn}$ values. This again demonstrates that size alone does not necessarily determines an animal's trophic level. We added a sentence on the size increase

and dental morphological similarities to our discussion (with a citation on a new article on *Galeocerdo* evolution, Türtscher et al., 2021; lines 359-361).

3.6

297-8: I'm not sure if this helps, but I wonder how much of this has to do with middle Miocene mysticetes being relatively less abundant than Pliocene mysticetes (e.g. mid Miocene assemblages are dominated by odontocetes with rare mysticetes).

Thank you very much for this comment. This is indeed true and is one of the reasons for our interpretation of Pliocene *O. megalodon* and *C. carcharias* feeding on mysticetes. In fact, small to medium-sized mysticetes remains are absent in our Miocene sites but can be found in the North American Pliocene sites. Naturally and as discussed in length in numerous prior publications, these Pliocene mysticetes were a likely prey item of *O. megalodon*. Expanding on the absence of these small to medium-sized mysticetes today and in the early Miocene compared to the Pliocene clearly supports our interpretation on the $\delta^{66}\text{Zn}$ results for both *O. megalodon* and *C. carcharias*. We realised now that we should have expanded more on this in the discussion and have done so now in our revised manuscript (lines 427-450).

3.7

300: As written, this citation of Pimiento and Balk suggests that they proposed climate change as a driver of *O. megalodon* extinction: they tested this, did not find evidence supporting it, and highlighted instead a biotic driver. This critical analysis is what led us to identify *Carcharodon carcharias* as the most likely biotic driver (which led you all to undertake this study). I suggest more carefully constructing the sentence before this citation to make it clear that these authors excluded climate change as a likely driver.

Done. With the rearrangement of our discussion, this is no longer an issue.

Reviewer #4 (Remarks to the Author):

4.1

The manuscript include important information about the use of Zinc isotope to found the trophic changes in Neogene megatooth shark, *Otodus megalodon*, a fossil shark tooth enameloid. Also they take samples from recent sharks and fish to do their comparison with fossil sharks.

I have some questions about the manuscript:

We thank the reviewer for recognising the importance of our work and hope that we have satisfactorily responded to the reviewer's comments and concerns.

4.2

L146: In the manuscript you mention about the fish used in your results: *Sparus auratus* from pisciculture. This species feed on controlled kind of food, comparing with the species in the wild. Why to use a fish and not a shark in aquarium, which feed on specific diet?

While not elasmobranchs (the main target of our study), there are several advantages to studying the pisciculture *Sparus aurata*. The *S. aurata* individuals in the pisciculture were fed exclusively with artificial pellets, thus the $\delta^{66}\text{Zn}$ values of their diet are unlikely to vary much. Indeed, pisciculture *S. aurata* enameloid $\delta^{66}\text{Zn}$ values from different individuals are within measurement uncertainty of each other, strongly contrasting $\delta^{66}\text{Zn}$ values of two wild *S. aurata* individuals caught nearby on the Israeli coast.

In contrast, aquaria shark teeth were less suited to demonstrate the clear dietary signal with our data for two main reasons. First, aquaria shark teeth were collected from the tanks after tooth shedding during the natural replacement process and we thus cannot assign the teeth to specific individuals. Second, the aquaria shark teeth were fed with wild caught fish and cephalopods (and additives), meaning their food source is likely to show some heterogeneity in $\delta^{66}\text{Zn}$ values. Indeed, aquaria shark $\delta^{66}\text{Zn}$ values show some variability, which however, further demonstrates that food not water uptake is controlling enameloid $\delta^{66}\text{Zn}$ values, as the tank seawater $\delta^{66}\text{Zn}$ values are less likely to vary much (although this remains to be tested).

To clarify why we include the pisciculture *S. aurata* while discussing control fed animals and do not focus on aquaria sharks we added the information about aquaria sharks being fed with wild caught fish and cephalopods (lines 220-221).

4.3

L 166 What are the main prey for mako shark and white shark in the wild? to see differences on zinc isotopes by each species.

Generally, all sharks are opportunistic. Mako sharks feed on prey from fish to marine mammals and the same can be said for the great white shark. We have analysed too few individuals to discuss differences between both species in terms of their respective $\delta^{66}\text{Zn}$ values, but our data clearly confirm that both are high trophic level apex predators. In response to the reviewer's comment, we have added relevant references discussing the diet and the apex predator nature of both species (line 202).

4.4

L168-169 Would you explain more about the differences of zinc isotopoe on marine ecosystem and terrestrial ecosystem?

Done. We have now added typical terrestrial mammal enamel $\delta^{66}\text{Zn}$ values as well as a statement, that besides the larger number of trophic levels in the marine ecosystem, differences between terrestrial and marine baselines may also contribute to the lower values in marine compared to terrestrial ecosystems (lines 203-206).

4.5

L171 In the manuscript you mention about *Galeocerdo cuvier* as an offshore pelagic shark; however this shark species are more from coastal habitats feeding on turtles, seabird, etc

We agree. *Galeocerdo cuvier* is a generalist and feeds on a variety of prey. It also has an essentially global distribution and can be found in coastal waters or far offshore. We only wanted to convey, that compared to *Carcharhinus*, adult *Galeocerdo cuvier* will typically consume less demersal, benthic coastal prey which is supported by stomach content and carbon and mercury isotope data. We expanded on this in lines 173-175 in the original manuscript and in Supplementary Discussion 3.1 with additional references.

To clarify, we have changed the sentence to: “*Carcharhinus* enameloid $\delta^{66}\text{Zn}$ values are high relative to sharks with similar bulk $\delta^{15}\text{N}_{\text{coll}}$ values, which contrary to the here analysed *Carcharhinus* species more regularly consume pelagic prey offshore, oceanic and on the continental shelf (e.g., *Galeocerdo cuvier*)²¹”.

References

Akhtar, A. A. et al. A record of the $\delta^{44}/^{40}\text{Ca}$ and [Sr] of seawater over the last 100 million years from fossil elasmobranch tooth enamel. *Earth Planet. Sci. Lett.* **543**, 116354. (2020). <https://doi.org/10.1016/j.epsl.2020.116354>

Cloquet, C., Carignan, J., Lehmann, M. F., & Vanhaecke, F. Variation in the isotopic composition of zinc in the natural environment and the use of zinc isotopes in biogeosciences: a review. *Anal. Bioanal. Chem.* **390**, 451-463 (2008).

Compagno, L. J. V. *Sharks of the world. An annotated and illustrated catalogue of shark species known to date. Bullhead, mackerel and carpet sharks (Heterodontiformes, Lamniformes and Orectolobiformes)*. (FAO Sp. Cat. Fish. Purp., Rome, 2001).

Condamine, F.L., Romieu, J. & Guinot, G. Climate cooling and clade competition likely drove the decline of lamniform sharks. *Proc. Natl. Acad. Sci.* **116**, 20584-20590 (2019).

Dorado, S., Rooker, J. R., Wissel, B. & Quigg, A. Isotope baseline shifts in pelagic food webs of the Gulf of Mexico. *Mar. Ecol. Prog. Ser.* **464**, 37-49 (2012).

Froese, R. & Pauly, D. FishBase. World Wide Web electronic publication. <https://www.fishbase.se/search.php>, version 06/2021 (2021).

Jaouen, K., Szpak, P. & Richards, M. P. Zinc isotope ratios as indicators of diet and trophic level in arctic marine mammals. *PLoS ONE* **11**, (2016). <https://doi.org/10.1371/journal.pone.0152299>

Liao, W. H. et al. Zn isotope composition in the water column of the Northwestern Pacific Ocean: the importance of external sources. *Global Biogeochem. Cycles* **34**, e2019GB006379 (2020). <https://doi.org/10.1029/2019GB006379>

Martin, J. E., et al., Calcium isotopic evidence for vulnerable marine ecosystem structure prior to the K/Pg extinction. *Curr. Biol.* **27**, 1641-1644 (2017).

Martin, J. E., Tacail, T., Adnet, S., Girard, C. & Balter, V. Calcium isotopes reveal the trophic position of extant and fossil elasmobranchs. *Chem. Geol.* **415**, 118-125 (2015).

McCormack, J. et al. Zinc isotopes from archaeological bones provide reliable trophic level information for marine mammals. *Commun. Biol.* **4**, 683 (2021). <https://doi.org/10.1038/s42003-021-02212-z>

Santora, J.A., et al. Diverse integrated ecosystem approach overcomes pandemic-related fisheries monitoring challenges. *Nat Commun* **12**, 6492 (2021). <https://doi.org/10.1038/s41467-021-26484-5>

Sundell, K. & Björnsson, B.T., Kinetics of calcium fluxes across the intestinal mucosa of the marine teleost, *Gadus Morhua*, measured using an *in vitro* perfusion method. *J. Exp. Biol.* **140**, 171–186. (1988).

Swanson, H. K. et al. A new probabilistic method for quantifying n-dimensional ecological niches and niche overlap. *Ecology* **96**, 318-324. (2015).

Türtscher, J. et al., Evolution, diversity, and disparity of the tiger shark lineage *Galeocerdo* in deep time. *Paleobiology*, **47**, 574-590 (2021).

Vennemann, T. W., Hegner, E., Cliff, G. & Benz, G. W. Isotopic composition of recent shark teeth as a proxy for environmental conditions. *Geochim. Cosmochim. Acta* **65**, 1583-1599 (2001).

Reviewers' Comments:

Reviewer #1:

Remarks to the Author:

Review for NCOMMS-21-38599A

I was previously Reviewer #1.

In this revised version of the manuscript, the authors have thoroughly addressed my comments and have made great efforts to take them into account while revising the manuscript. I commend the authors for this work. I can only say that I have enjoyed reading the first version of the study, but I am even more convinced now with the revision and responses made to the questions I had.

I think this study will foster new research avenues in paleobiology, and as such I would support the publication of this study in Nature Communications.

Dr. Fabien L. Condamine

Reviewer #2:

Remarks to the Author:

The revision addressed one of my major comments - about the impossibility of linking the extinction of the mega-toothed shark to competition with the great white shark based on the results presented - by being more careful with the expression of the major aims and conclusions. They addressed questions about oceanic conditions in different parts of the world, including most completely, concerns about the influence of wind-blown dust. It's clearer now why the authors eschewed statistical representations of the data encompassing several isotopes in combination given the small numbers from multiple places in the fossil record in particular. However, a difficulty remains for the reader in 'finding one's way through' a large plethora of data for species extant and extinct, from multiple places and several fossil periods, and the multiple statistical comparisons. This was one reason for the earlier suggestion to represent and test several isotopes in combination. The authors argued in response that, for many samples, there are too few from each location to use explore the interactions of carbon, nitrogen and Zn isotopes. This is a good point but it does then raise a question about the choices of samples. This is of course a problem in most studies in palaeontology so understandable, but far less so for the modern sample choices.

For people who do not understand the particular regional names for periods of the Miocene and Pliocene I suggest adding age depictions that are more generic (not everyone understands the timing of Burdigalian for instance) in the text and in the sample list. Having looked it up (=Early Miocene ca 17-24Ma if memory serves), the age gaps in the ages of fossils becomes clearer.

Also, regarding the oxygen isotopes, suggest that they forefront the more cogent point that they reflect water column conditions of temperature and salinity (as expressed in Line 317-318 in the edited copy of the manuscript). The idea to explore diagenesis in relation to oxygen isotope composition is not very convincing given that their explorations of zinc isotopes and concentrations seem nicely illuminating as regards diagenesis (or lack of it).

Reviewer #3:

Remarks to the Author:

I'm fairly satisfied with the author's responses to my queries. To be clear - I was **definitely not** suggesting the reviewers conduct a quantitative study of bite marks- however, if they have an enterprising student, it would be a great master's or PhD thesis.

However, regarding other edits - Reviewer 1 is NOT correct that *Orcinus* is known from the Zanclean

(early Pliocene). There is an isolated tooth of *Orcinus* from the Setana Formation reported by Kohno and Tomida (1993: Bulletin Nat. Sci. Mus. Tokyo, Ser. C 19:4:139-146). The Setana Formation is early Pleistocene (it's right there in the abstract of the paper). The only pre-Pliocene fossil of *Orcinus* known is *Orcinus citoniensis*, which is a relatively small pilot whale like odontocete with a higher count of smaller teeth, and it was not a macrophagous predator; several papers by G. Bianucci discuss this (e.g. Bianucci 1996, 2005). Further, Boessenecker et al. (2019: PeerJ) thoroughly disputed any competition between *C. megalodon* and *Orcinus* - the former going extinct before the latter evolved large body size and macrophagous habits. [It's further worth noting that other studies (e.g. Marsili 2008 - Atti. Soc. tosc. Sci. Nat. Mem. Serie A, 113:81-88) already indicate that the Mediterranean fossil record of *C. megalodon* terminates in the Zanclean, and thus locally *Orcinus* and *C. megalodon* are even stratigraphically separated).

Bianucci, G. 1996. The Odontoceti (Mammalia, Cetacea) from Italian Pliocene. Systematics and Phylogeny of Delphinidae. *Palaeontographia Italica* 83:73–167.

Bianucci, G. 2005. *Arimidelphis sorbinii* a new small killer whale-like dolphin from the Pliocene of Marecchia river (Central eastern Italy) and a phylogenetic analysis of the Orcininae (Cetacea: Odontoceti). *Rivista Italiana di Paleontologia e Stratigrafia* 111:329–344.

Reviewer #4:

None

Reviewer #5:

Remarks to the Author:

I don't have too many remarks. The paper is sound.

The arguments against diagenetic alteration are reasonable, at least to the degree of having interpretable results.

It may be that there is some variability in zinc isotopic baselines (either between regions or over time) but the authors make a good argument that this is probably minimal (in that some taxa seem not to vary in zinc isotopes).

I found Supplementary Figure 12 useful when reading the paragraphs beginning at line 278 and line 311. If there is room could this be moved to the main text?

Author response

We are thankful for the constructive feedback given by all reviewers. In the following, we discuss the comments made by each reviewer (original remarks are in black, our response in blue). When referring to changes made with line numbers, we refer to the tracked changes versions of the manuscript unless otherwise specified.

Reviewer #1 (Remarks to the Author):

Review for NCOMMS-21-38599A

I was previously Reviewer #1.

In this revised version of the manuscript, the authors have thoroughly addressed my comments and have made great efforts to take them into account while revising the manuscript. I commend the authors for this work. I can only say that I have enjoyed reading the first version of the study, but I am even more convinced now with the revision and responses made to the questions I had.

I think this study will foster new research avenues in paleobiology, and as such I would support the publication of this study in Nature Communications.

Dr. Fabien L. Condamine

We thank the reviewer for his kind words and appreciation of our work.

Reviewer #2 (Remarks to the Author):

The revision addressed one of my major comments - about the impossibility of linking the extinction of the mega-toothed shark to competition with the great white shark based on the results presented - by being more careful with the expression of the major aims and conclusions. They addressed questions about oceanic conditions in different parts of the world, including most completely, concerns about the influence of wind-blown dust.

We thank the reviewer for the insightful review, and we are glad that we were able to address these major comments to the reviewer's satisfaction.

It's clearer now why the authors eschewed statistical representations of the data encompassing several isotopes in combination given the small numbers from multiple places in the fossil record in particular. However, a difficulty remains for the reader in 'finding one's way through' a large plethora of data for species extant and extinct, from multiple places and several fossil periods, and the multiple statistical comparisons. This was one reason for the earlier suggestion to represent and test several isotopes in combination. The authors argued in response that, for many samples, there are too few from each location to use explore the interactions of carbon, nitrogen and Zn isotopes. This is a good point but it

does then raise a question about the choices of samples. This is of course a problem in most studies in palaeontology so understandable, but far less so for the modern sample choices.

We appreciate that the reviewer agrees that our changes to the manuscript clarified the statistical representation of our data. We indeed present an unusually large and completely new isotope dataset (392 individual $\delta^{66}\text{Zn}$, $\delta^{15}\text{N}_{\text{coll}}$, $\delta^{13}\text{C}_{\text{coll}}$, $\delta^{18}\text{O}_\text{P}$ values) covering many extant and fossil taxa (with differing diets, geographical habitats, and ages). However, this is inherently necessary to demonstrate the feasibility of using this novel proxy in both extant fish tissues and deep-time ecological studies.

Our sample set was chosen specifically for the goal of demonstrating the variability and reliability of $\delta^{66}\text{Zn}$ values as a dietary/trophic proxy in modern and fossil fish enameloid. Importantly, the various isotope analyses conducted (oxygen, carbon, and nitrogen) and the multitude of sites and periods studied (for Zn) served specific purposes of either controlling for Zn variability (i.e., baselines [spatial and temporal], pelagic vs. benthic, etc.) or validating the Zn data (i.e., diagenesis, trophic levels, etc.). In fact, having extant individuals from different geographic locations is actually an advantage here (e.g., discussing potential baseline variations). We should also stress that, again, for this pioneering zinc isotope study, we were not able to combine a large range of extant shark teeth from a single location due to the many strict protection laws and regulations, and therefore difficulties in gaining access to modern material, particularly for destructive/consumptive sampling necessary for our study. Besides, establishing multi-isotope niches for extant sharks was not the objective of this pioneering study. Subsequent studies can and will further explore the interactions of carbon, nitrogen, and zinc isotopes in modern ecosystems; such work will undoubtedly bring about tremendous insights for both an ecological perspective and the interpretative framework of Zn variability. We thus consider our large dataset and its representation in the text and figures to be assets for the scientific community.

For people who do not understand the particular regional names for periods of the Miocene and Pliocene I suggest adding age depictions that are more generic (not everyone understands the timing of Burdigalian for instance) in the text and in the sample list. Having looked it up (=Early Miocene ca 17-24Ma if memory serves), the age gaps in the ages of fossils becomes clearer.

We only refer to the more detailed (and lesser-known) chronostratigraphic stages, such as the Burdigalian, in the Supplementary Information when providing a geological context to the samples (1.2 Supplementary sample information). In the main text and Supplementary Discussion, we use more commonly known geological epoch names (Early Miocene, Miocene-Pliocene transition, Pliocene). Nevertheless, we now include dates when referring to epoch names for the first time in the text (lines 163-164).

Also, regarding the oxygen isotopes, suggest that they forefront the more cogent point that they reflect water column conditions of temperature and salinity (as expressed in Line 317-318 in the edited copy of the manuscript). The idea to explore diagenesis in relation to oxygen isotope composition is not very convincing given that their explorations of zinc isotopes and concentrations seem nicely illuminating as regards diagenesis (or lack of it).

We agree that our discussion about fossil zinc isotopes and concentrations is sufficient to demonstrate the lack of diagenetic modification of pristine zinc isotope values in enameloid. This is why oxygen isotopes

do not play a major role in the diagenesis discussion; in fact, they are mentioned only in one sentence of the diagenesis discussion (line 175-178 original manuscript). Nevertheless, we have changed the introduction (lines 71-73) to not give the impression that $\delta^{18}\text{O}_p$ values were used solely to explore diagenesis.

Reviewer #3 (Remarks to the Author):

I'm fairly satisfied with the author's responses to my queries. To be clear - I was *definitely not* suggesting the reviewers conduct a quantitative study of bite marks- however, if they have an enterprising student, it would be a great master's or PhD thesis.

We are glad that the reviewer is pleased with our responses, and we wish to thank the reviewer for reviewing and appreciating our work.

However, regarding other edits - Reviewer 1 is NOT correct that *Orcinus* is known from the Zanclean (early Pliocene). There is an isolated tooth of *Orcinus* from the Setana Formation reported by Kohno and Tomida (1993: Bulletin Nat. Sci. Mus. Tokyo, Ser. C 19:4:139-146). The Setana Formation is early Pleistocene (it's right there in the abstract of the paper). The only pre-Pliocene fossil of *Orcinus* known is *Orcinus citoniensis*, which is a relatively small pilot whale like odontocete with a higher count of smaller teeth, and it was not a macrophagous predator; several papers by G. Bianucci discuss this (e.g. Bianucci 1996, 2005). Further, Boessenecker et al. (2019: PeerJ) thoroughly disputed any competition between *C. megalodon* and *Orcinus* - the former going extinct before the latter evolved large body size and macrophagous habits. [It's further worth noting that other studies (e.g. Marsili 2008 - Atti. Soc. tosc. Sci. Nat. Mem. Serie A, 113:81-88) already indicate that the Mediterranean fossil record of *C. megalodon* terminates in the Zanclean, and thus locally *Orcinus* and *C. megalodon* are even stratigraphically separated).

Bianucci, G. 1996. The Odontoceti (Mammalia, Cetacea) from Italian Pliocene. Systematics and Phylogeny of Delphinidae. *Palaeontographia Italica* 83:73–167.

Bianucci, G. 2005. *Arimidelphis sorbinii* a new small killer whale-like dolphin from the Pliocene of Marecchia river (Central eastern Italy) and a phylogenetic analysis of the Orcininae (Cetacea: Odontoceti). *Rivista Italiana di Paleontologia e Stratigrafia* 111:329–344.

As reviewers 1 and 3 disagree on this topic, we omitted including the name *Orcinus* when discussing potential other resource competitors of *Otodus megalodon*, while still not excluding the potential competition with carnivorous odontoceti in general. The sentence in the final paragraph (lines 313-316) now reads:

“The extinction of *O. megalodon* could have been caused by multiple, compounding environmental and ecological factors^{46,47}, including climate change and thermal limitations⁴⁸, the collapse of prey populations⁴ and resource competition with *C. carcharias*¹⁵ and possibly other taxa not examined here (e.g., carnivorous odontoceti).”

This way, we believe we can please both reviewers. In any case, as no odontoceti teeth were examined in this study, this issue does not hold much relevance here.

Reviewer #5 (Remarks to the Author):

I don't have too many remarks. The paper is sound.

The arguments against diagenetic alteration are reasonable, at least to the degree of having interpretable results.

It may be that there is some variability in zinc isotopic baselines (either between regions or over time) but the authors make a good argument that this is probably minimal (in that some taxa seem not to vary in zinc isotopes).

Thank you very much.

I found Supplementary Figure 12 useful when reading the paragraphs beginning at line 278 and line 311. If there is room could this be moved to the main text?

We agree that this figure is very helpful for this discussion. We initially decided against including the figure in the main text to avoid a repetition of the data representation. All data points in Supplementary Figure 12 are already represented in Figures 2 and 3, albeit represented differently with a different focus. Following the reviewers suggestion, we now include Supplementary Figure 12 as Figure 4 in the main text.